# A contextual fear conditioning paradigm in head-fixed mice exploring virtual reality

**Seetha Krishnan\*[†], Can Dong[†‡], Heather Ratigan, Denisse Morales-Rodriguez[§], Chery Cherian, Mark Sheffield\***

Department of Neurobiology and Institute for Neuroscience, University of Chicago, Chicago, United States

## eLife Assessment

This **useful** study presents a virtual reality-based contextual fear conditioning paradigm for head-fixed mice. **Solid** evidence supports the claim that the reported methods provide a reliable paradigm for studying contextual fear conditioning in head-fixed mice. The approach provides a way to perform multiphoton imaging of neural circuits, and other techniques that are typically performed in head-fixed animals, during behaviors that have traditionally been studied in freely moving animals.

**Abstract** Contextual fear conditioning (CFC) is a classical laboratory task that tests associative memory formation and recall. Techniques such as multi-photon microscopy and holographic stimulation offer tremendous opportunities to understand the neural underpinnings of these memories. However, these techniques generally require animals to be head-fixed. Few paradigms examine contextual fear in head-fixed mice, and none use freezing—the most common measure of fear in freely moving animals—as the behavioral readout. To address this gap, we developed a CFC paradigm for head-fixed mice using virtual reality (VR). We designed an apparatus to deliver tail shocks while mice navigated a VR environment. We tested three versions of this paradigm and, in all of them, observed increased freezing, particularly on the first trial, in the shock-paired VR compared to a neutral one. These results demonstrate that head-fixed mice can be fear-conditioned in VR and exhibit context-specific freezing behavior. Additionally, using two-photon calcium imaging, we tracked large populations of hippocampal CA1 neurons before, during, and following CFC. As in freely moving mice, CA1 place cells remapped and developed narrower fields following fear conditioning. Thus, our approach enables new opportunities to study the neural mechanisms underlying the formation, recall, and extinction of contextual fear memories.

## Introduction

Large-scale multi-photon imaging and holographic stimulation offers the ability to accurately track and record from the same neuronal populations over extended periods at high spatiotemporal resolution, record from axons, dendrites, and dendritic spines, and stimulate neurons with single-cell precision (*Bando et al., 2021*; *Chaigneau et al., 2016*; *Dombeck et al., 2010*; *Dong et al., 2021*; *Emiliani et al., 2015*; *Fan et al., 2023*; *Hattori and Komiyama, 2022*; *Heer and Sheffield, 2024*; *Hernandez et al., 2016*; *Krishnan et al., 2022*; *Russell et al., 2022*; *Sheffield and Dombeck, 2015*). These techniques facilitate the study of neural circuit dynamics, synaptic and plasticity mechanisms, and enable the real-time manipulation of neural activity, all of which are crucial for understanding the mechanisms underlying memory formation and recall in the brain. However, head-fixation of animals,

**\*For correspondence:**
seethakrishnan@uchicago.edu (SK);
sheffield@uchicago.edu (MS)

[†]These authors contributed equally to this work

**Present address:** [‡]Department of Neurobiology, Stanford University, Palo Alto, United States; [§]University of California, San Francisco, San Francisco, United States

**Competing interest:** The authors declare that no competing interests exist.

which is generally required to utilize these techniques, imposes constraints on the range of testable behaviors compared to freely moving conditions (*Guo et al., 2014*; *Juczewski et al., 2020*). This limitation is particularly evident in the Pavlovian CFC task, a laboratory method for testing associative learning, fear memory formation, and recall (*Bouton, 1993*; *Fanselow, 1980*; *Fanselow, 1990*; *Kim and Fanselow, 1992*; *LeDoux, 2000*; *Maren, 2001*; *Maren et al., 2013*).

In the traditional CFC paradigm for freely moving animals, an animal is placed in an environment (conditioned stimulus, CS) that is paired with an aversive stimulus, such as a mild foot shock (unconditioned stimulus, US). Animals are then removed from the environment. When reintroduced to the environment, animals exhibit a species-specific conditioned response, such as fearful freezing behavior in rodents, if they successfully associate the shock-paired environment with the aversive stimulus. The CFC paradigm is one of the most basic conditioning procedures for freely moving animals— it's simple and robust (*Fanselow, 1980*; *Fanselow, 1990*). Despite its longstanding use in studies involving freely behaving animals to explore questions on learning and memory (*Anagnostaras et al., 2001*; *Bouton and King, 1983*; *Bouton and Bolles, 1980*; *Debiec et al., 2002*; *Fanselow and Bolles, 1979*; *Fendt and Fanselow, 1999*; *Gewirtz et al., 2000*; *Holland and Bouton, 1999*; *Kim and Jung, 2006*; *Maren et al., 2013*; *Maren and Holt, 2000*; *Pezze and Feldon, 2004*), it has been noticeably absent in studies involving head-fixed animals, with some exceptions.

Two prior studies (*Lovett-Barron et al., 2014*; *Rajasethupathy et al., 2015*) reported a version of CFC in head-fixed mice with aversive air puffs as the US. These studies reported the conditioned response animals displayed as lick suppression, not freezing. Although lick suppression after fear conditioning has also been observed in freely moving animals (*Leaf and Leaf, 1966*; *Leaf and Muller, 1965*), it has been shown to result from increased freezing (*Bouton and Bolles, 1980*). Therefore, relying solely on lick suppression in the head-fixed preparation makes it more challenging to compare with the extensive previous work in freely moving animals, where freezing is measured as the predominant conditioned fear response. Furthermore, the aforementioned head-fixed CFC studies also had the limitation of simultaneously having both a water reward and an aversive stimulus within the context, potentially confounding the results and complicating the interpretation. Hence, we aimed to develop a paradigm where freezing is the conditioned response and without the presence of a reward within the conditioned context, making it directly comparable to freely moving animals.

Here, we describe three different versions of a contextual fear-conditioning paradigm for head-fixed mice that resulted in freezing as the conditioned response. The context or CS was a virtual reality (VR)-based environment, which head-fixed mice explored by running on a cylindrical treadmill. In a previous study (*Ratigan et al., 2023*), we demonstrated the feasibility of this approach by combining it with imaging of thalamic axons innervating the hippocampus using a two-photon microscope. The present manuscript provides a comprehensive description and fundamentals of the behavioral framework, to enable the research community to reproduce and adapt our behavior paradigm. Our main aim is to show that VR-based CFC can elicit freezing responses similar to those seen in freely moving animals, while also examining how key parameters affect this response. Though we explored some important variables, this paper does not attempt to comprehensively investigate all parameters necessary for successful VR-based CFC. Additionally, we performed two-photon imaging of a large population of hippocampal CA1 cells during fear conditioning and tracked the same cells during memory recall across days. We analyzed the impact of our paradigm on place cells in the hippocampus and compared these findings to results from CFC in freely moving animals.

## Results

### A paradigm for contextual fear conditioning in head-fixed mice exploring virtual reality environments

To establish the contextual fear conditioning protocol (*Figure 1A–C*), head-fixed mice were water-restricted and trained to run on a treadmill and navigate VR environments for water rewards, as we have demonstrated previously (*Dong et al., 2021*; *Krishnan et al., 2022*; *Krishnan and Sheffield, 2023*; *Ratigan et al., 2023*). Forward movement on the treadmill translates to forward movement in VR, allowing mice to navigate the VR environment. The environments used were 2m-long linear tracks rich in visual cues (*Figure 1*). Reaching the track's end triggered a water reward and defined a completed trial/lap. Mice were then virtually teleported back to the track's start for the next lap.

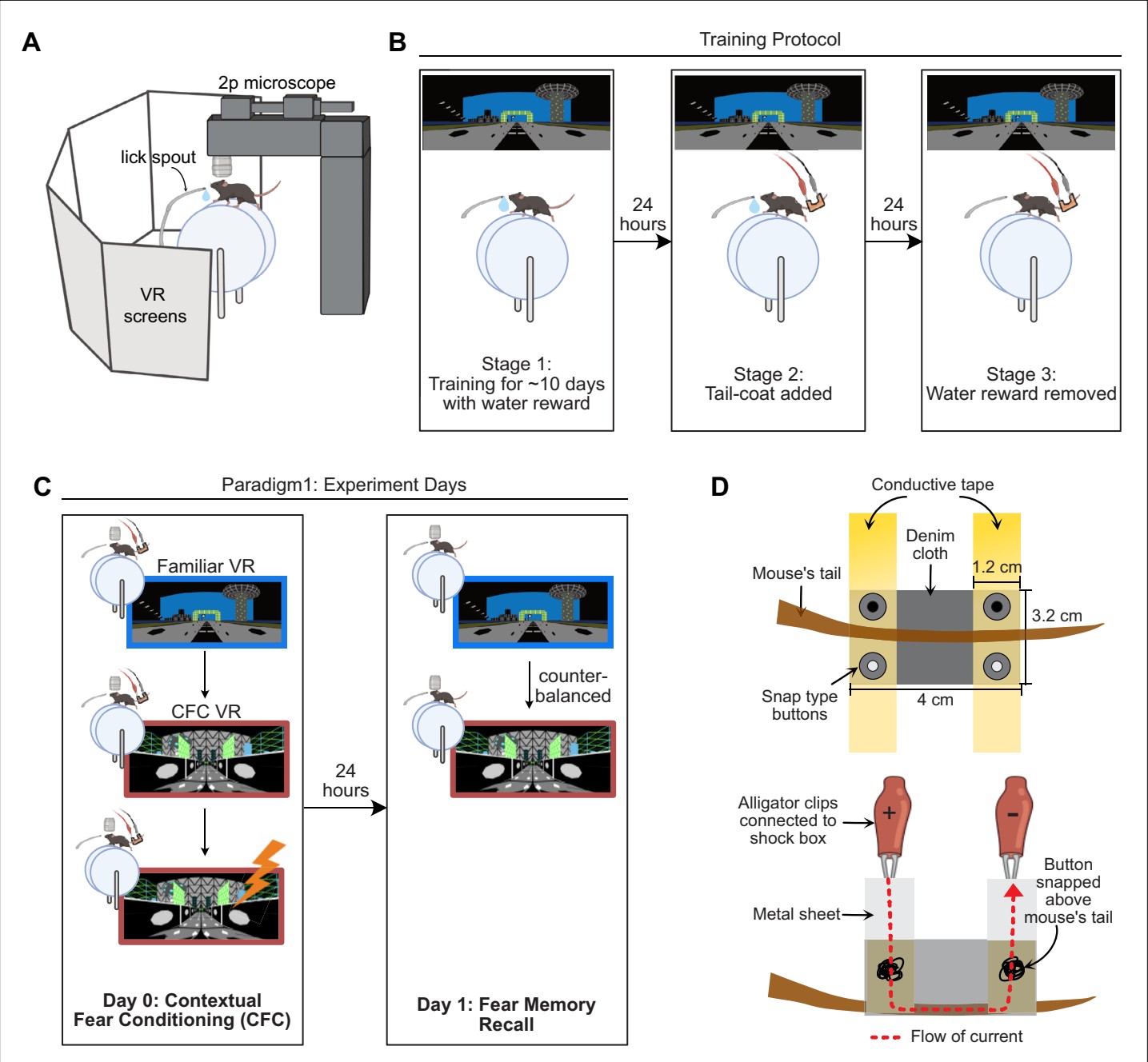

**Figure 1.** A contextual fear conditioning paradigm for head-fixed mice navigating virtual reality environments. (**A**) Experimental setup created with BioRender.com. Mice were head-restrained with their feet resting on a cylindrical treadmill. Five large monitors surrounded the mice that displayed virtual reality (VR) environments. Movement on the treadmill advanced the VR display, allowing for context exploration. (**B**) Mice were water-restricted and trained to run laps in the VR for water rewards. VR environments were 2m-long linear tracks. Mice were trained to achieve >3 laps per minute, which took ~10–14 d (Stage 1). Once well-trained, a 'tail-coat' was added to their tails (Stage 2), followed by the removal of the water reward the next day (Stage 3). (**C**) For Paradigm 1, mice underwent the fear conditioning protocol the following day after the water reward was removed. On the first experiment day (Day 0), mice spent 10 min in the training VR (Familiar VR) and then another 10 min in a new VR (contextual fear conditioning, CFC VR). After the initial exploration, mice received mild electric shocks on the tail (4–12 shocks, 0.5–1.2mA in amplitude, 1 s long). The next day (Day 1), mice were tested for memory recall by placing them in the Familiar VR and the CFC VR for 5 min each in a counterbalanced manner. (**D**) Schematic of the tail-coat used for delivering mild electric shocks to the mouse's tail. (Top) View from the top (Bottom) side view. Dimensions are provided for a typical 12-wk-old male mouse weighing ~30 g before water restriction (see Methods for more details).

Unlike freely moving mice that tend to explore new environments, head-fixed mice placed on a treadmill do not automatically start exploring VR environments. However, they can be trained through reinforcement learning using water restriction and water rewards (*Dombeck et al., 2010*; *Dong et al., 2021*; *Krishnan et al., 2022*; *Ratigan et al., 2023*; *Sheffield and Dombeck, 2015*).

Training to navigate VR environments on a treadmill (Stage 1), using water rewards, was achieved in approximately 10–14 d, after which well-trained mice could run several laps per minute. We set a criterion of >3 laps per minute (i.e. a lap velocity of 0.1–0.2 m/s) for mice to move on to the next stage (Stage 2). Once the mice reached this stage (Stage 2), the following day, we added a coat to their tails (*Figure 1B*). We developed a lightweight, wearable conductive apparatus for the mouse's tail to ensure minimal discomfort for the mouse. We named this apparatus the 'tail-coat' (*Figure 1D*). This tail-coat enabled us to administer mild electric shocks to the mouse's tail (see Methods).

When the tail-coat was added, some mice reduced their running speed or refused to move. Some recovered the next day, but if their running speed did not reach at least three laps per minute (~20% of mice failed to meet this criterion), they were not advanced to the next stage. Mice that maintained this running speed advanced to Stage 3. Data from mice that advanced with minimal changes to their lap running behavior after adding the tail-coat are shown in *Figure 2—figure supplement 1A*.

The following day (Stage 3), mice were placed in the same VR environment with the tail-coat, but the water reward was removed (*Figure 1B*). Mice that continued to run at least three laps per minute with the tail-coat and without water reward advanced to the experiment stage (Stage 4). Around 40% of the mice dropped out at this stage. High levels of lap running during training were crucial to ensure mice were undeterred by both the tail-coat and the reward removal (*Krishnan et al., 2022*; *Krishnan and Sheffield, 2023*). Our criteria for consistent running behavior with minimal pauses throughout these three stages of training ensured that any baseline freezing behavior was minimal, similar to freely moving animals, and allowed us to attribute freezing after CFC to fear conditioning.

In summary, our criteria for selecting mice for the experimental stage of contextual fear conditioning required them to run consistently (>3 laps per minute) with (1) rewards, then (2) with a tail-coat, and then (3) without rewards plus the tail-coat. This threshold resulted in about 40% of the mice proceeding to the experimental stage. Once mice qualified for the experimental stage, they were no longer excluded from analysis due to their behavior.

We tested three versions of the head-fixed CFC paradigm. In Paradigm 1, the neutral environment was the training VR environment, and a new VR environment was paired with shocks. In Paradigms 2 and 3, both the neutral and shock-paired environments were new and distinct from the training environment to better control for environmental familiarity. In Paradigms 1 and 2, the tail-coat was removed on the days following CFC, whereas in Paradigm 3, it remained on the mouse on all post-CFC days. With these paradigms, we varied some parameters that may influence fear memory recall, contextual discrimination, fear extinction, and their associated behaviors in head-fixed VR-based CFC paradigms. We detail these variations in the following sections, starting with Paradigm 1.

## When tested for contextual fear memory recall, mice displayed increased freezing behavior in the shocked-paired VR environment

On Day 0 of the experimental stage in Paradigm 1 (CFC day), mice (n=27, 25 males and 2 females) were exposed to the same VR environment we used for training, referred to as the Familiar VR from here onwards (*Figure 1C*), for 10 min. We then switched them to a new VR environment (CFC VR). Following baseline exploration of the CFC VR for 10 min, we repeatedly applied mild-electric shocks (4–12 shocks, 0.5-1 mA in amplitude, 1 s long) to the mouse's tail. We call the period when shocks were delivered 'during CFC.' Shocks were pseudo-randomly administered with respect to the animal's position in the CFC VR, with an inter-shock interval (ISI) of 1 min. To ensure that the shocks were associated with the context and not a specific position in the environment, no visual cues were explicitly paired with the shock. We quantified the animal's response to the shocks and observed an immediate increase in running speed following each shock (*Figure 2—figure supplement 1B*). This was consistent across all mice and shocks and was a reliable indicator that the apparatus was working correctly. Sixty seconds after the last shock was administered, the VR screens were switched off, the tail-coat was removed, and mice were returned to their home cages. The next day (Fear memory recall, Day 1), we tested for recall of the learned association between the VR environment and the fearful stimulus. Mice were exposed to both VRs (familiar VR and CFC VR) for 5 min each in a counterbalanced manner.

Water reward remained absent on recall days; however, in this paradigm, the tail-coat was no longer present.

Before fear conditioning, mice exhibited similar behavior in both the Familiar and CFC VRs. We evaluated the time taken to complete a lap and the freezing behavior (defined as time points when the animal's instantaneous velocity was 0 cm/s). Neither metric revealed a significant difference between the two VRs before shock administration (*Figure 2A*, *Figure 2—figure supplement 1D-E*). However, after CFC, when the animals were reintroduced to the VRs on recall Day 1, we observed a significant increase in freezing and time taken to complete a lap in the CFC VR compared to the Familiar VR (*Figure 2A–D*). As soon as mice transitioned to the CFC VR (either from dark or after the Familiar VR) on the recall day, we observed behaviors such as slowing down, freezing, moving backward, and hesitation to move forward (*Videos 1–3*). This response was markedly different from their behavior in the Familiar VR. This difference in behavior can be observed clearly in most mice on the very first lap (*Figure 2A–B*). Since we removed the reward, we didn't find licking behavior to be indicative of recall behavior (*Figure 2—figure supplement 1C*). However, some mice displayed licking behavior in the Familiar VR on Day 0 before fear conditioning, which was absent on Day 1. This is likely because mice had previously received a water reward in the Familiar VR. We will address this caveat in Paradigms 2 and 3.

To quantify the distinct conditioned responses observed (*Figure 2A–B*, *Videos 1–3*), we used two metrics: freezing behavior, defined as periods when the animal's instantaneous velocity was 0 cm/s, and time taken to complete a lap, which captured the avoidance-like behaviors (such as slowing down, moving backward, and hesitation to move forward) as well as freezing. When we quantified the freezing behavior in mice, we found it to be, on average, high throughout the 5 min that mice spent in the CFC VR on recall Day 1 (*Figure 2C–D*, average freezing (%), mean ± 95% Confidence Intervals (CI), Familiar VR: 17.52±3.25, CFC VR: 23.54±3.26, p=0.001, Paired *t*-test, n=27). We also observed a significant increase in freezing during the first lap in the CFC VR compared to the familiar VR (*Figure 2B–C*, average first lap freezing (%), Familiar VR: 15.94±4.03, CFC VR: 25.11±3.41, p<0.001, Paired *t*-test). Mice also took longer to complete a lap in the CFC VR compared to the Familiar VR, both in the first lap (average time taken to complete first lap (s), Familiar VR: 14.81±3.74, CFC VR: 41.16±9.33, p<0.001, Paired *t*-test) and on average (*Figure 2E–F*, the average time taken to complete all laps (s), Familiar VR: 16.17±3.05, CFC VR: 38.82±11.06, p<0.001, Paired *t*-test). The increase in freezing and time taken to complete a lap in the CFC VR compared to the Familiar VR was significantly higher on Day 1 than Day 0, indicating that it results from fear conditioning (*Figure 2C–F*, p<0.05, Paired *t*-test).

Based on studies in freely moving animals (*Fanselow, 1980*; *Fanselow, 1990*; *Ramanathan et al., 2018*; *Russo and Parsons, 2021*; *Trott et al., 2022*), we settled on an intensity of 1 mA and six shocks as a reasonable protocol, which we tested on a sizeable cohort of mice (n=12). We found that freezing at this intensity and shock number was higher on average in the CFC VR; however, mice showed variability in their responses (*Figure 2—figure supplement 2A–B*). The majority of mice (7/12) displayed appropriate memory recall (*Figure 2—figure supplement 2A–B*), indicated by more freezing in the CFC VR compared to the Familiar VR (delta >0), while 3/12 mice froze more in the Familiar VR (delta <0), and 2/12 showed equal freezing levels in both VRs (delta ≈ 0). Such variability has been regularly documented in freely moving animals and may not be unique to the VR setup used here (*Chu et al., 2024*; *Dos Santos Corrêa et al., 2019*; *Navarro-Sánchez et al., 2024*; *Poulos et al., 2016*; *Russo and Parsons, 2021*; *Totty et al., 2021*).

Next, in a small group of mice (n=15), we investigated whether varying the number of shocks or the shock amplitude would affect the conditioned fear response (*Figure 2—figure supplement 2A–B*). Though our parameter exploration was not exhaustive, we assessed shock amplitude more extensively than the number of shocks. (*Figure 2—figure supplement 2C–E*). Overall, most shock amplitudes and numbers of shocks increased freezing behavior in the CFC VR on recall day (*Figure 2—figure supplement 2A–B*). The observed variability was within the same range as that of the larger cohort of mice we tested using the 1 mA, 6-shock protocol (*Figure 2—figure supplement 2A–B*). Notably, increasing the shock amplitude from 0.6 to 1 mA, while keeping the number of shocks consistent at 6 shocks, significantly enhanced both fear discrimination and freezing behavior (*Figure 2—figure supplement 2C–E*). A more detailed analysis with a larger sample size may be necessary to understand further how variations in shock parameters affect the conditioned response, as has been done

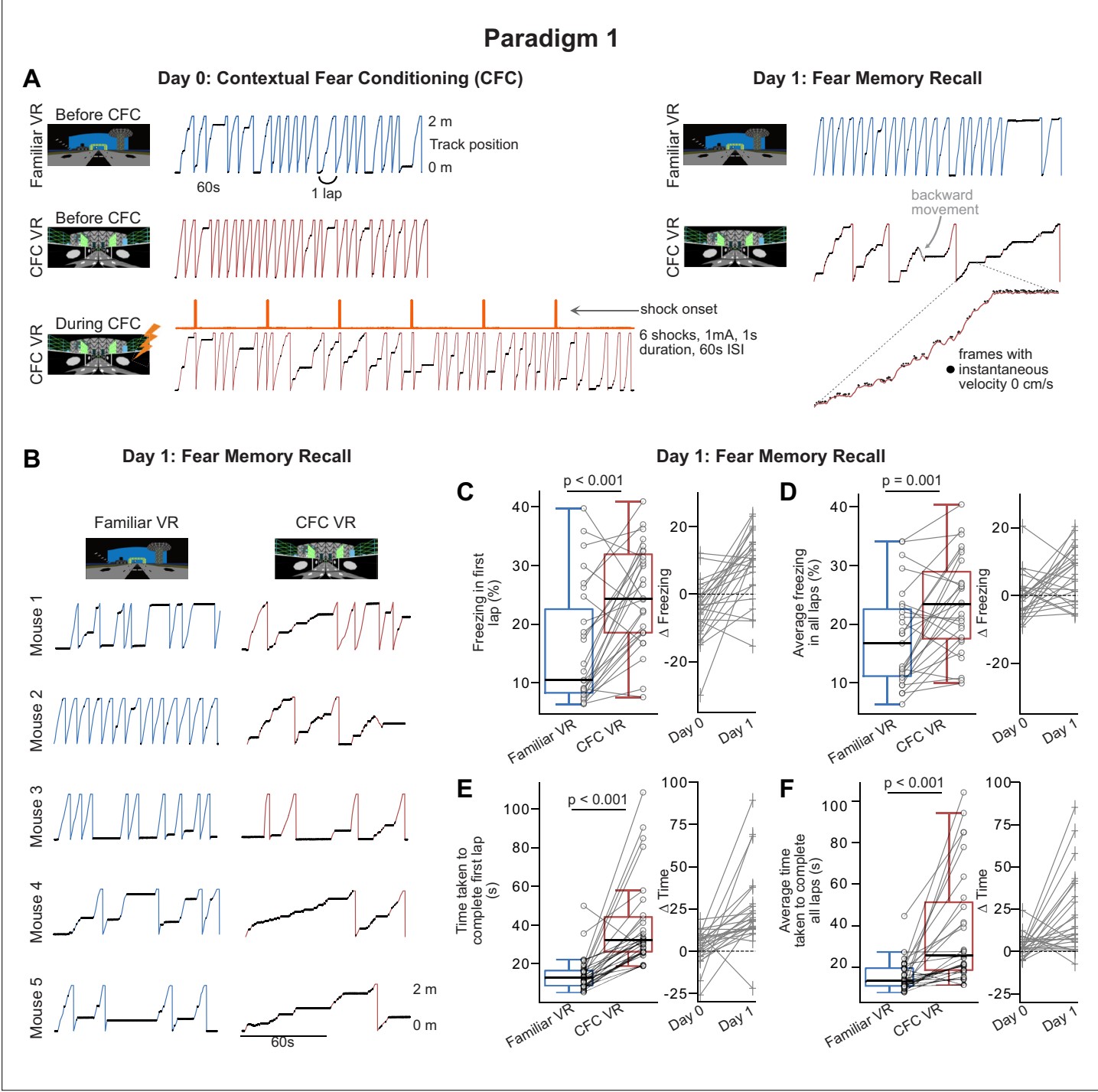

**Figure 2.** Paradigm 1: Head-fixed mice show increased freezing behavior following contextual fear conditioning in the contextual fear conditioning (CFC) virtual reality (VR). (**A**) A single example mouse's lap running behavior on experiment days, Day 0 and Day 1. Behavior is shown for approximately 3 min in all sessions except during CFC, which is shown for the 6 min that the session lasted. Frames where freezing was detected (instantaneous velocity 0 cm/s) are marked with black dots. A zoomed-in portion on the right in CFC VR highlights these freezing epochs as periods of minimal movement on the treadmill (i.e. instantaneous velocity 0 cm/s), which are less visible in the full-scale view. The traces on the right show that this mouse increased freezing, decreased velocity, and moved backward (shown in gray) in the CFC VR (red traces) but not in the Familiar VR (blue traces) on Recall Day. This mouse received six shocks at 1mA intensity at 60 s inter-stimulus interval (ISI). (**B**) First 2 min of recall behavior in more mice (n=5) in Familiar VR versus CFC VR. (**C–D**) Left Average freezing percentage on recall day in the very first lap (**C**) and all laps (**D**) during the 5 min that mice explored the Familiar (blue) and CFC (red) VR. Freezing (%) was calculated as the number of frames where freezing was detected in a lap divided by the total number of frames in each lap. (Right) Delta calculated as the difference in the amount of freezing in the CFC VR compared to the Familiar VR before CFC (Day

*Figure 2 continued on next page*

*Figure 2 continued*

0) and after CFC (Day 1). The dashed line represents 0. (**E–F**) Same as C-D but for the time taken to complete the first lap (**E**) and all laps (**F**). Mice displayed an increase in freezing and in time taken to complete a lap in the first lap and, on average, in the CFC VR. In C-F, circles and pluses represent individual mice (n=27, 25 male, and two female mice). In C-F, data was pooled from mice receiving different numbers of shocks (4, 6, 12) at varying intensities (0.5 mA, 0.6mA, 0.8 mA, 1 mA, and 1.2 mA), which is separately displayed in *Figure 2—figure supplement 1*. Lines join data from the same mouse. p-values were calculated using a paired t-test.

The online version of this article includes the following figure supplement(s) for figure 2:

**Figure supplement 1.** Running and licking behaviors in Paradigm 1.

**Figure supplement 2.** Change in freezing with shock amplitude, number of shocks, and across recall days in Paradigm 1.

in studies with freely moving animals (*Baldi et al., 2004*; *Cordero et al., 1998*; *Daviu et al., 2024*; *Poulos et al., 2016*; *Quinn et al., 2008*).

Finally, we examined the evolution of this conditioned response across days (*Figure 2—figure supplement 2F–I*). We placed the animals in both the Familiar and the CFC VR for 5 min over 4 d (Recall Days 1–4) in a counterbalanced manner and tested their recall behavior. We quantified both the degree of fear discrimination across days (*Figure 2—figure supplement 2F–G*) and the extinction of the fear response within the CFC VR compared to baseline (*Figure 2—figure supplement 2H–I*).

Fear discrimination was most prominent on Recall Day 1. Mice displayed the most substantial increase in freezing in the CFC VR compared to the Familiar VR on Recall Day 1 (*Figure 2—figure supplement 2F–G*). Freezing on the first lap in the CFC VR was also most pronounced on Recall Day 1. Both average and first lap freezing in the CFC VR decreased on Recall Day 2 compared to Recall Day 1, becoming similar to the Familiar VR. While first lap freezing remained similar between the Familiar and the CFC VR on subsequent Recall Days 3 and 4, we observed some variability in the average freezing behavior. Average freezing increased significantly again on Recall Day 3 in the CFC VR but was similar in both VRs on Recall Day 4.

To determine the extinction of the fear response, we compared freezing within each VR to baseline freezing levels on Day 0 (*Figure 2—figure supplement 2H–I*). We quantified this using a Linear Mixed Effects (LME) Model, treating recall days as fixed effects and mouse as a random effect, with baseline freezing on Day 0 as the reference (*Tables 1–2*). In the CFC VR, both first lap and average freezing were most prominent on Recall Day 1 and returned to baseline levels by Recall Day 2. While we observed a small increase in average freezing on Recall Day 3, it remained lower than Recall Day 1 levels and returned to baseline by Recall Day 4. This pattern suggests that the pronounced first lap freezing observed on Recall Day 1 extinguishes quickly, while average freezing, though more variable, also extinguishes by Recall Day 4.

In the Familiar VR, freezing behavior—both first lap and average—did not differ significantly from baseline on most days. However, we observed an unexpected increase in average freezing compared to baseline on Recall Day 4. This increase could reflect either fear generalization or a decrease in motivation - a response to the prolonged absence of reward in a previously rewarded environment.

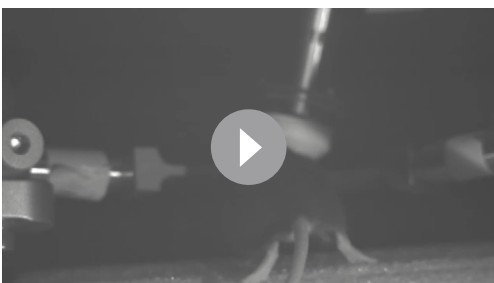

**Video 1.** The video shows the behavior of a mouse on Recall Day 1 in the neutral environment that was not paired with shocks. Video was recorded immediately after the mouse transitioned to the virtual reality (VR) environment, and they were collected at a sampling rate of 30 frames/s.

https://elifesciences.org/articles/105422/figures#video1

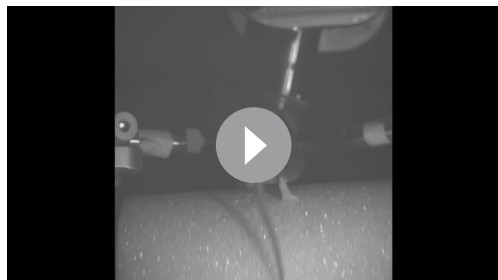

**Video 2.** The video shows the freezing behavior of a mouse on Recall Day 1 in the contextual fear conditioning (CFC) virtual reality (VR). Video was recorded immediately after the mouse transitioned to the VR environment, and they were collected at a sampling rate of 30 frames/s.

https://elifesciences.org/articles/105422/figures#video2

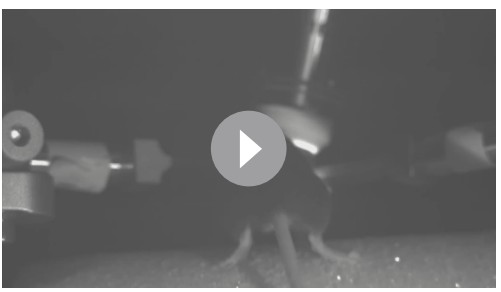

**Video 3.** The video shows a mouse's behavior on Recall Day 1 in the contextual fear conditioning (CFC) virtual reality (VR). This mouse displays more avoidance-like behaviors, such as slowing down and making backward movements. Video was recorded immediately after the mouse transitioned to the VR environment, and they were collected at a sampling rate of 30 frames/s. https://elifesciences.org/articles/105422/figures#video3

To distinguish between these possibilities, we examined freezing behavior in control mice that underwent the same paradigm without shocks (no-shock controls, n=7, black lines in *Figure 2—figure supplement 2H–I*). Average freezing in the Familiar VR exceeded that of no-shock controls on both Recall Days 1 and 4. Since no-shock controls explored the environments under no-reward conditions, this suggests that the increase in freezing in the Familiar VR stems from fear conditioning and may indicate generalization rather than motivational changes. This aligns with findings from freely moving animals, where specific fear memories have been shown to generalize over time with repeated exposures to both neutral and conditioned contexts (*Biedenkapp and Rudy, 2007*; *Wiltgen and Silva, 2007*).

In summary, these results demonstrate that head-fixed mice can acquire fear associations in VR environments. Most mice displayed freezing as a conditioned response specifically to the shock-paired VR and showed robust fear discrimination—behavior that mirrors that of freely moving animals after fear conditioning. By the fourth recall day, freezing had largely extinguished. Notably, beyond freezing, other avoidance-like defensive behaviors were prominent in the shock-paired VR, demonstrating the range of behavioral responses our paradigm can capture.

## A second paradigm where head-fixed mice discriminate between two novel VRs

As previously mentioned, the incomplete extinction of licking behavior in the familiar VR before fear conditioning or the gradual increases in freezing behavior in the familiar VR during recall tests may indicate that mice still associate the VR with water rewards, which could be a caveat. Furthermore, in traditional CFC paradigms with freely moving mice, fear behavior has been assessed mainly by comparing two novel contexts, one associated with shocks and the other acting as a control. To address this, we tested another paradigm (Paradigm 2), using a novel VR instead of the familiar VR to act as the control (*Figure 3A*). One novel VR was assigned as the Control VR, and a second novel VR was assigned as the CFC VR, which would later become paired with the shocks. In this paradigm, we also added an extra habituation day to increase the animal's pre-exposure time to the novel VRs prior to conditioning (Day–1). Pre-exposure to the context before fear conditioning can enhance contextual fear, as seen in freely moving animals (*Biedenkapp and Rudy, 2007*; *Fanselow, 1980*; *Fanselow, 1990*; *Maren et al., 2013*; *Rudy et al., 2002*; *Wiltgen and Silva, 2007*).

The training was done similarly to the previous paradigm. After mice showed consistent running behavior in the Familiar VR with the tail-coat and without reward, they were advanced to the experimental stage. On Day –1, mice were exposed to the two novel VRs for 10 min each. On Day 0, they were re-exposed to the same two VRs for 10 min before receiving six 1-s-long mild electric shocks (at 1 mA and 60 s ISI) in one of them - the CFC VR. On Day 1, mice were exposed to the two VRs for 5 min to evaluate memory recall (*Figure 3A–B*).

**Table 1.** Linear mixed effect (LME) summary (estimate ± standard error, p-value) for first lap freezing in Paradigm 1 with fixed effects (recall days, baseline: before CFC, Day 0) and random effect (mouse).

| VR | Recall Day 1 | Recall Day 2 | Recall Day 3 | Recall Day 4 |
|---|---|---|---|---|
| Familiar VR | –0.97±1.95, p=0.62 | –5.75±2.51, p=0.02 | –4.13±3.03, p=0.18 | 1.08±3.18, p=0.73 |
| CFC VR | 12.96±2.02, p<0.001 | 3.42±2.55, p=0.18 | 1.73±3.07, p=0.57 | 1.26±3.23, p=0.70 |

**Table 2.** Linear mixed effect (LME) summary (estimate ± standard error, p-value) for average freezing in Paradigm 1 with fixed effects (recall days, baseline: before CFC, Day 0) and random effect (mouse).

| VR | Recall Day 1 | Recall Day 2 | Recall Day 3 | Recall Day 4 |
|---|---|---|---|---|
| Familiar VR | 1.00±1.13, p=0.37 | −0.43±1.48, p=0.77 | 0.76±1.80, p=0.67 | 4.87±1.88, p=0.013 |
| CFC VR | 7.55±1.58, p<0.001 | 2.89±2.04, p=0.16 | 6.03±2.47, p=0.018 | 3.78±2.58, p=0.15 |

Compared to Paradigm 1, in the novel VRs, mice showed no differences in licking behavior across all experimental sessions (*Figure 3—figure supplement 1A*). There was also no significant difference in freezing behavior and time taken to complete a lap between the two VRs on Day −1 and Day 0 before fear conditioning (*Figure 3—figure supplement 1B–C*).

## Head-fixed mice show heightened fear discrimination during the first lap when discriminating between two novel VRs

We compared the freezing behavior between Paradigm 1 and Paradigm 2 on Recall Day 1 to the same intensity and number of shocks (1 mA, 1 s long, 60 s ISI, six shocks, n=12 mice each paradigm). While first lap freezing in the CFC VR was markedly higher in Paradigm 2, significant differences between the Control VR and CFC VR were observed only for first lap freezing, not for average freezing or lap completion time across all laps (*Figures 3B–D and 4A–E*). The effect sizes for the difference in first lap freezing between the two VRs were nearly identical between the paradigms (Cohen's d: First lap freezing: Paradigm 1: 1.038, Paradigm 2: 1.040), but the effect size for average freezing was notably smaller in Paradigm 2 compared to Paradigm 1 (Cohen's d: Average Freezing: Paradigm 1: 0.797, Paradigm 2: 0.093).

When we quantified freezing by lap on Recall Day 1 in Paradigm 2, we found that after the first lap, mice showed similar freezing in both the Control VR and the CFC VR across all subsequent laps (*Figure 4—figure supplement 1*). In contrast, in Paradigm 1, mice consistently froze more in the CFC VR compared to the Familiar VR throughout Recall Day 1 (*Figure 4—figure supplement 1*).

Indeed, when we quantified freezing behavior in the Control VR and CFC VR in Paradigm 2, we found that the most significant increase in freezing and other avoidance-like defensive behaviors in the CFC VR was observed in the first lap (freezing in first lap %: Control VR: 11.04±1.8 CFC VR: 16.05±3.94, p=0.016, time taken to complete first lap (s): Control VR: 14.27±2.85, CFC VR: 25.68±5.55, p=0.001, Paired *t*-test, *Figure 3B–E*, *Figure 4A–E*). Neither average freezing across all laps (*Figure 4C*, average freezing in all laps %: Control VR: 20.54±4.13, CFC VR: 21.15±4.21, p=0.738, Paired *t*-test) nor time taken to complete all laps (*Figure 4E*, the average time taken to complete all laps: Control VR: 21.73±8.11, CFC VR: 24.27±6.36, p=0.558, Paired *t*-test) showed significant differences between the two VRs.

When mice were returned to the two VRs the following day for a second day of recall (Recall Day 2), they continued to exhibit modest levels of fear discrimination, as indicated by a small but statistically significant difference in freezing in the first lap and on average between the CFC VR and Control VR (*Figure 4—figure supplement 2A–B*). However, when assessing fear extinction by comparing freezing in the CFC VR across recall days to baseline levels (*Figure 4—figure supplement 2C–D*, *Tables 3–4*), we found no significant differences beyond the very first lap on Recall Day 1. This suggests that while mice maintained their ability to discriminate between environments by freezing more in the CFC VR, their overall fear response had extinguished. Thus, in Paradigm 2, fear extinction occurs rapidly within the first recall session after the initial lap.

## A shorter ISI and leaving the tail-coat on during fear memory recall improved fear discrimination when using two novel VRs

Freezing behavior in the CFC VR in Paradigm 2 was weaker than in Paradigm 1, with rapid fear extinction after the first recall lap. In Paradigm 3, two parameters were changed relative to Paradigm 2 to test whether freezing behavior and fear discrimination could be enhanced. First, the interval between consecutive shocks was reduced to see if a shorter ISI could produce a more robust fear response (*McNally and Westbrook, 2006*; *Williams, 1994*). Second, the tail-coat was kept on during memory recall, based on evidence that removing contextual cues can reduce conditioned freezing (*González*

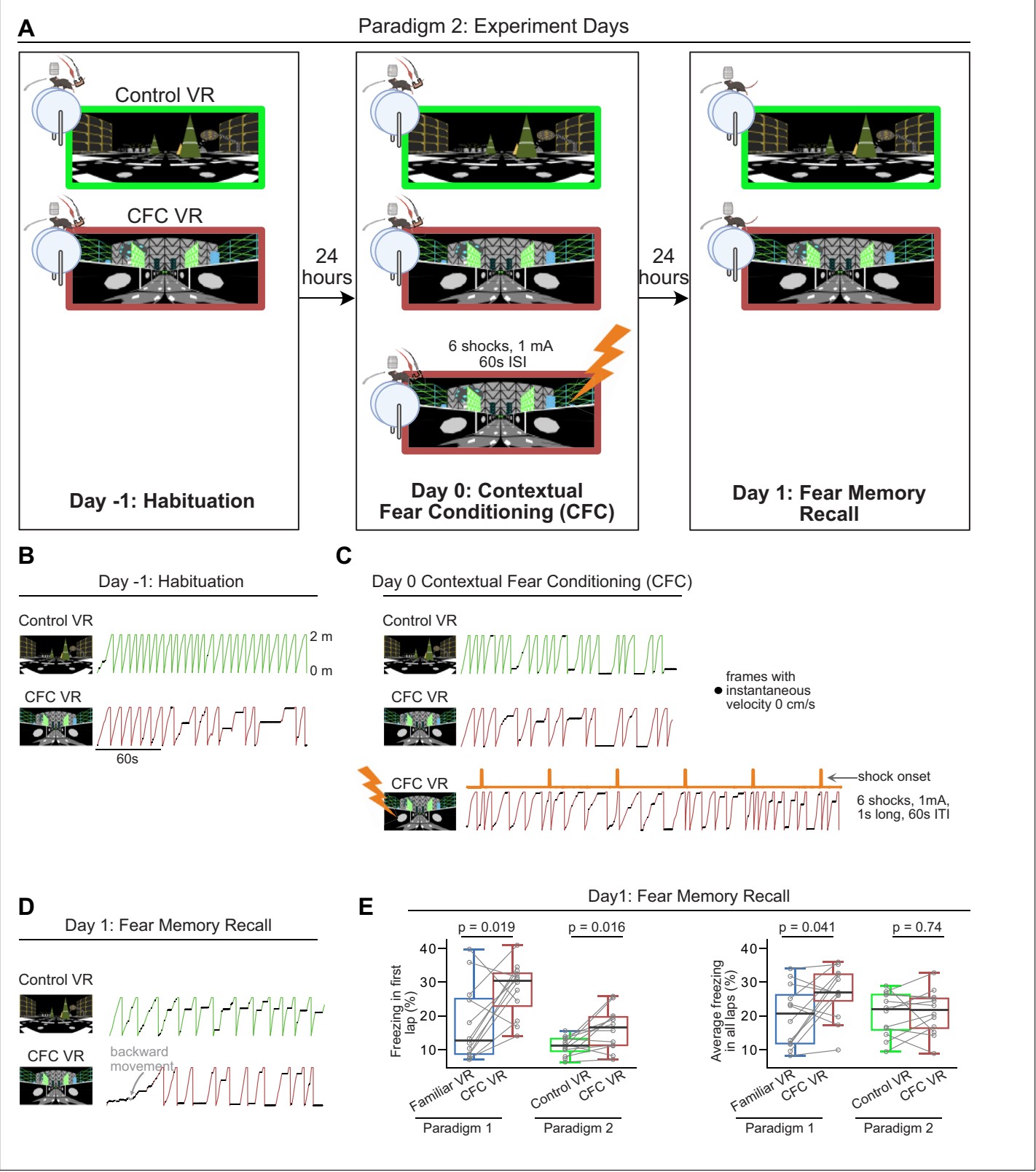

**Figure 3.** Using a novel virtual reality (VR) as the neutral environment instead of a familiar VR in a second paradigm results in modest increases in freezing. (**A**) The training paradigm for Paradigm 2 is similar to Paradigm 1. On experiment day, mice were introduced to two novel VRs, one of which would be associated with the shock (contextual fear conditioning, CFC VR) and the other wouldn't (Control VR). In this paradigm, there was an added habituation day, where mice were exposed to the two VRs for 10 min. The next day, mice ran in the two VRs again before receiving mild electric shocks

*Figure 3 continued*

in the CFC VR. The recall test occurred the next day. In this cohort, all mice received six shocks at 1 mA intensity. (**B–D**) These panels show the lap running behavior of a single mouse on experiment days –1, 0, and 1. Behavior is displayed for about 3 min in all sessions except during CFC. Frames with freezing detected by a threshold (instantaneous velocity 0 cm/s) are marked by black dots. In (**D**), this mouse shows an increase in freezing, a decrease in velocity, and some backward movement when in the CFC VR (red traces). (**E**) Comparison between freezing in the first lap (left) and all laps (right) in Paradigm 1 (blue and red) versus Paradigm 2 (green and red). In Paradigm 1, only animals that received six shocks at 1 mA intensity are included. n=12 mice were used in both paradigms (10 male and two female mice). The scale bar for panels B-D is indicated in B. p-values were calculated using a paired t-test.

The online version of this article includes the following figure supplement(s) for figure 3:

**Figure supplement 1.** Licking and running behavior before fear conditioning, in Paradigm 2.

---

*et al., 2003*). We hypothesized that the tail-coat might act as a contextual cue and that its presence during recall might enhance freezing behavior. To further test these effects, we used a lower shock intensity of 0.6 mA—which we previously showed as producing only modest fear responses in Paradigm 1 (*Figure 2—figure supplement 2C–E*)—to determine whether a shorter ISI and the retention

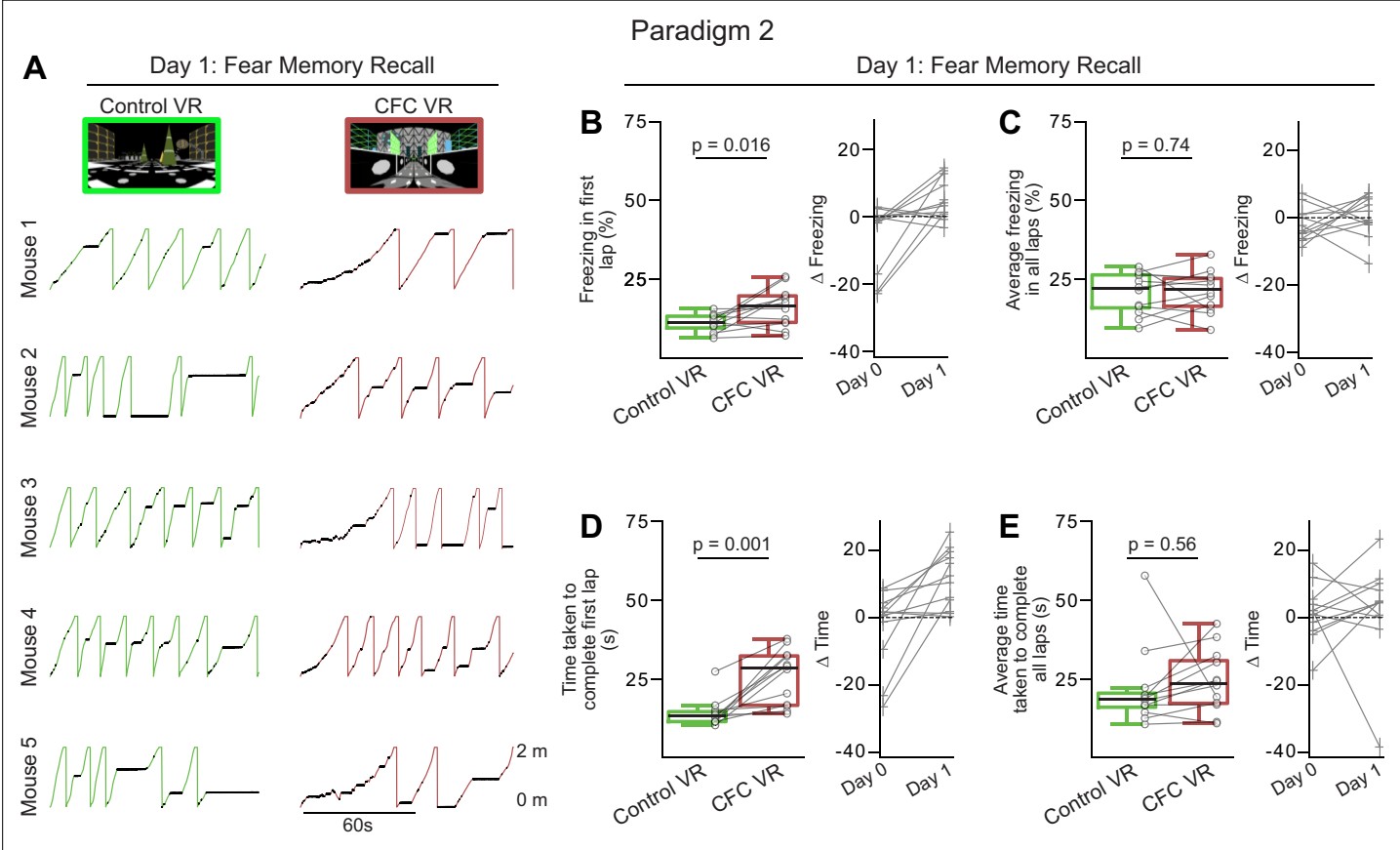

**Figure 4.** In Paradigm 2, the most significant freezing and reduced speed occurred in the first lap of the contextual fear conditioning (CFC) virtual reality (VR) compared to the Control VR. (**A**) First 2 min of recall behavior in five mice in Control VR (green) versus CFC VR (red). (**B–C**) The left panels show the amount of freezing on the recall day in (**B**) the very first lap and (**C**) all laps during the 5 min when the mice explored the Control (green) and CFC (red) VR. Freezing (%) is calculated as the number of frames where freezing was detected in a lap divided by the total number of frames in each lap. On the right, the delta is calculated as the difference in the amount of freezing in the CFC VR compared to the Control VR before (Day 0) and after CFC (Day 1). The dashed line represents 0. (**D–E**) Same as B-C but for the time taken to complete the first lap (**D**) and all laps (**E**). In B-E, circles and pluses represent individual mice (n=12). Lines join data from the same mouse. p-values were calculated using a paired t-test.

The online version of this article includes the following figure supplement(s) for figure 4:

**Figure supplement 1.** Lap-wise freezing behavior in the three paradigms.

**Figure supplement 2.** Mice largely extinguished their fear within the first day of recall in Paradigm 2.

**Table 3.** Linear mixed effect (LME) summary (estimate ± standard error, p-value) for first lap freezing in Paradigm 2 with fixed effects (recall days, baseline: before CFC, Day 0) and random effect (mouse).

| VR | Recall Day 1 | Recall Day 2 |
|---|---|---|
| Control VR | −5.22±2.88, p=0.08 | −6.99±3.22, p=0.04 |
| CFC VR | 5.02±1.98, p=0.02 | 4.01±2.24, p=0.09 |

of the tail-coat during recall could enhance conditioned fear responses produced by the lower shock intensity.

In Paradigm 3, mice followed the same experimental protocol as Paradigm 2, with two novel VRs: one assigned as the Control VR and the other as the CFC VR. A habituation day in the two new VRs (Day −1) was followed by the CFC day (Day 0), where mice received six shocks (0.6 mA, 20–26 s ISI) in the CFC VR. This was followed by recall days (Days 1–3), where mice explored the two VRs presented in a counter-balanced manner. Unlike the other two paradigms, in Paradigm 3, the tail-coat remained on during the recall days (*Figure 5A*).

We found that mice (n=20 male mice) showed better fear discrimination between the Control and CFC VRs in Paradigm 3 compared to Paradigm 2 (*Figure 5B–D*). Mice not only showed significant freezing in the first lap (freezing in first lap %: Control VR: 13.61±2.24 CFC VR: 19.73±5.17, p=0.021, Paired *t*-test) but average freezing across all laps was also higher in the CFC VR compared to the Control VR (*Figure 5C–D*, *Figure 6A–D*, average freezing in all laps %: Control VR: 15.28±2.00, CFC VR: 21.92±2.95, p<0.001, Paired *t*-test).

While first lap freezing showed a weaker effect size in Paradigm 3 compared to the other paradigms, there was a strong effect of average freezing across the session (Cohen's d: First lap freezing: Paradigm 1: 1.038, Paradigm 2: 1.040, Paradigm 3: 0.719. Average Freezing: Paradigm 1: 0.797, Paradigm 2: 0.093, Paradigm 3: 1.232). On average, we found that Paradigm 1 still produced the highest absolute freezing behavior in the CFC VR (average freezing in all laps %: CFC VR: Paradigm 1: 23.54±3.26, Paradigm 2: 21.51±4.21, Paradigm 3: 21.92±2.95). The difference in freezing behavior between the CFC VR and the neutral VR was comparable between Paradigms 1 and 3 (Δ average freezing % (CFC VR – Familiar/Control VR): Paradigm 1: 6.74, Paradigm 2: 0.61, Paradigm 3: 6.63). Thus, unlike Paradigm 2, where the largest differences in freezing between the CFC VR and the Control VR were restricted to the first lap, the shorter ISIs and the addition of the tail-coat during the recall day in Paradigm 3 improved contextual fear discrimination between the two novel VRs. When tested for memory recall in Paradigm 3, mice froze more in the CFC VR than in the Control VR, and this effect persisted beyond the first lap (*Figure 5C–D*, *Figure 4—figure supplement 1*). Mice also maintained their ability to discriminate between the two VRs for the first two recall days, freezing significantly more in the CFC VR than in the Control VR on both Recall Days 1 and 2 (*Figure 6—figure supplement 1A–B*). However, by Recall Day 3, this discrimination diminished, as freezing levels no longer differed significantly between the two VRs.

Unlike the rapid fear extinction observed in Paradigm 2, mice in Paradigm 3 showed delayed extinction (*Tables 5–6*, *Figure 6—figure supplement 1C–D*). Freezing levels in the CFC VR remained elevated above baseline during the first lap and across all laps on Recall Days 1 and 2, with average freezing still elevated above baseline on Recall Day 3. While freezing levels showed a consistent downward trend across days, these results suggest that fear had not completely extinguished by Recall Day 3. As in Paradigm 1, the most pronounced increases in first lap freezing in the CFC VR occurred in early recall days, while elevated average freezing persisted longer. These findings suggest

**Table 4.** Linear mixed effect (LME) summary (estimate ± standard error, p-value) for average freezing in Paradigm 2 with fixed effects (recall days, baseline: before CFC, Day 0) and random effect (mouse).

| VR | Recall Day 1 | Recall Day 2 |
|---|---|---|
| Control VR | −2.71±2.00, p=0.19 | −1.15±2.27, p=0.62 |
| CFC VR | −0.23±2.06, p=0.91 | 2.76±2.32, p=0.25 |

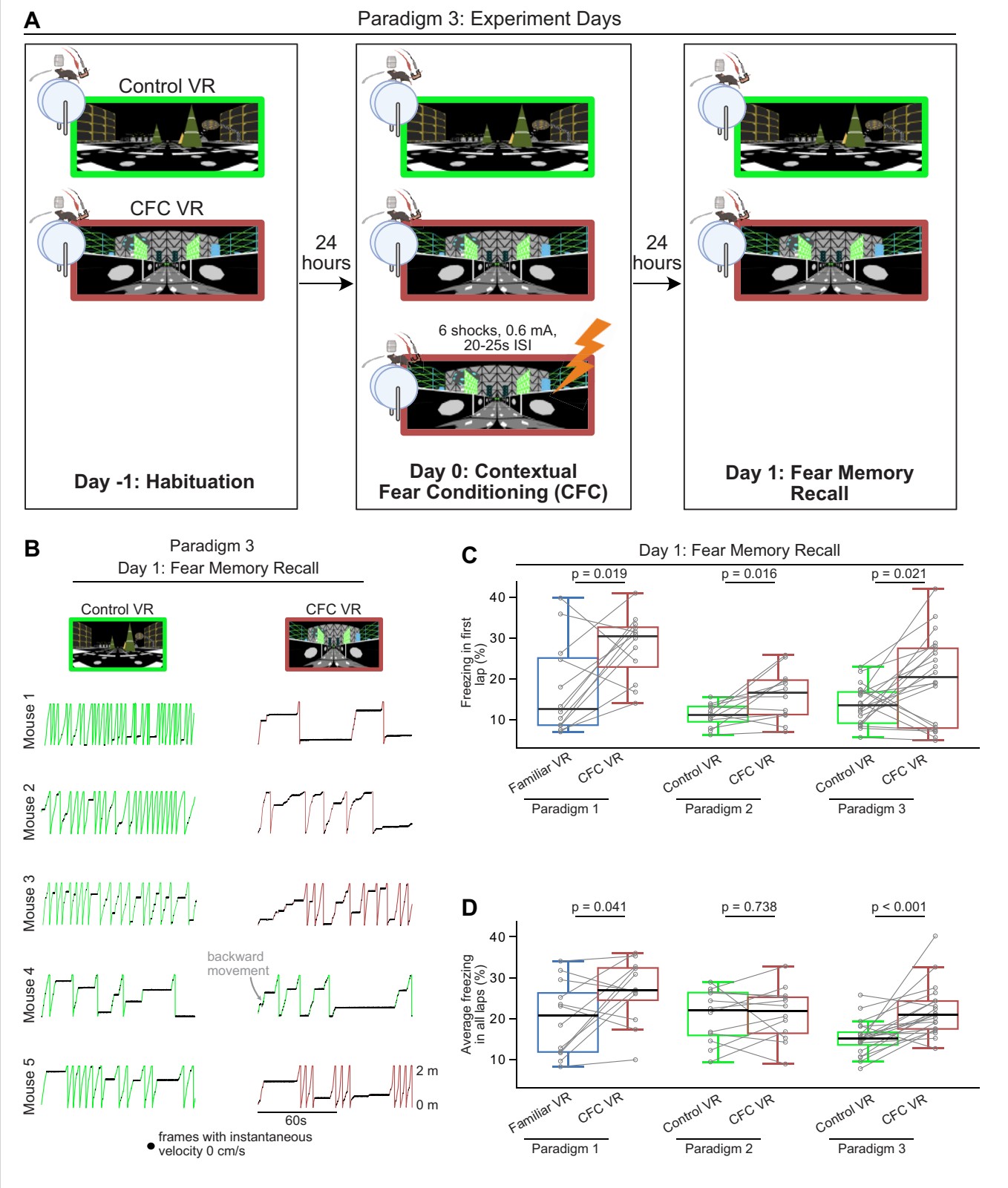

**Figure 5.** A third paradigm that uses a novel virtual reality (VR) as the neutral environment but keeps the tail-coat on during memory recall led to increased freezing in the contextual fear conditioning (CFC) Environment. (**A**) The training and experiment paradigm was similar to Paradigm 2 except that the shocks were administered closer together (20–25 s ISI), and the tail-coat was kept on during recall days. (**B**) First 2 min of recall behavior in Paradigm 3 in five mice in Control VR (green) versus CFC VR (red). (**C–D**) Comparison between freezing in the first lap (**C**) and all laps (**D**) in Paradigm

*Figure 5 continued*

1 (blue and red) versus Paradigms 2 and 3 (green and red). n=12 mice were used in Paradigms 1 and 2 (10 male and two female mice), and n=20 male mice in Paradigm 3. p-values were calculated using a paired t-test. In Paradigm 3, by keeping the tail-coat on during the recall days, we observed an increase in freezing in the CFC VR in the first lap and across all laps compared to Paradigm 2.

that the shorter ISI and presence of the tail-coat during memory recall may enhance conditioned fear and delay fear extinction in head-fixed mice discriminating between two novel VRs, though the individual effects of these parameters require further investigation.

The tail-coat's presence during recall days likely contributed to some transfer of fear to the Control VR, evidenced by elevated freezing in the Control VR compared to baseline. Nevertheless, freezing

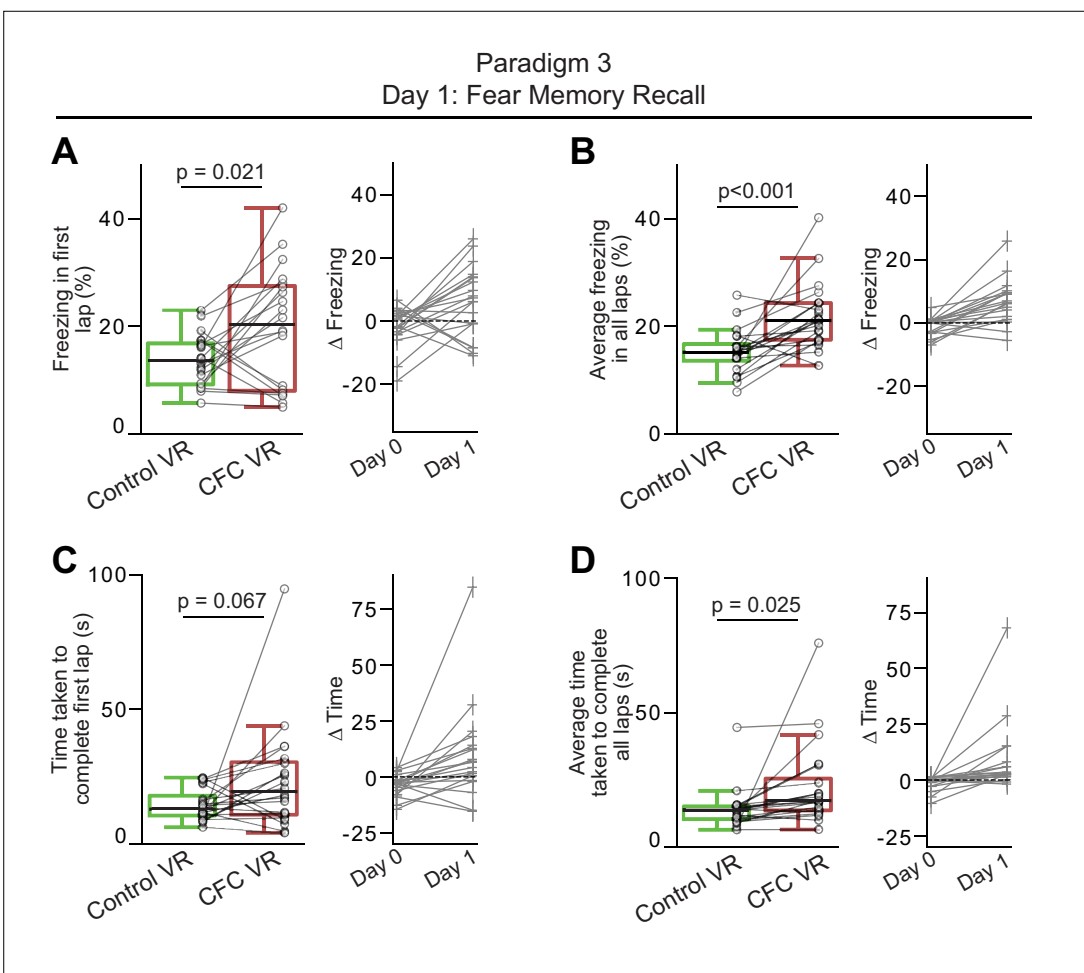

**Figure 6.** Mice exhibited increased freezing and overall reduced speed in the contextual fear conditioning (CFC) virtual reality (VR) in Paradigm 3. (**A–B**) The left panels show the amount of freezing on the recall day in (**A**) the very first lap and (**B**) all laps during the 5 min when the mice explored the Control (green) and CFC (red) VR. Freezing (%) is calculated as the number of frames where freezing was detected in a lap divided by the total number of frames in each lap. On the right, the delta is calculated as the difference in the amount of freezing in the CFC VR compared to the Control VR before (Day 0) and after CFC (Day 1). The dashed line represents 0. (**C–D**) Same as A-B but for time taken to complete the first lap (**C**) and all laps (**D**). Circles and pluses represent individual mice (n=20). Lines join data from the same mouse. p-values were calculated using a paired t-test.

The online version of this article includes the following figure supplement(s) for figure 6:

**Figure supplement 1.** Mice displayed better fear discrimination and a delay in extinction in Paradigm 3 compared to Paradigms 1 and 2.

**Table 5.** Linear mixed effect (LME) summary (estimate ±standard error, p-value) for first lap freezing in Paradigm 3 with fixed effects (recall days, baseline: before CFC, Day 0) and random effect (mouse).

| VR | Recall Day 1 | Recall Day 2 | Recall Day 3 |
|---|---|---|---|
| Control VR | 1.84±1.92, p=0.34 | 3.01±1.92, p=0.12 | −2.09±1.92, p=0.28 |
| CFC VR | 9.67±2.61, p=0.0005 | 6.84±2.61, p=0.01 | 3.33±2.61, p=0.21 |

levels in the Control VR remained consistently lower than in the CFC VR across all days (*Figure 6—figure supplement 1*, *Tables 5–6*).

In summary, our results show that mice exhibited greater average freezing behavior in the CFC VR in Paradigms 1 and 3 compared to Paradigm 2. In Paradigm 2, the most noticeable increase in freezing occurred during the very first lap in the CFC VR compared to the Control VR. Fear extinction was rapid in Paradigm 2 and was back to baseline levels within the first day of recall. We found that fear discrimination could be improved, and fear extinction could be delayed with a shorter ISI when presenting tail shocks and when the tail-coat was present during memory recall, as shown in Paradigm 3. Thus, we provide three different variations for performing CFC in head-fixed mice that can be used to understand the neural underpinnings of fear memory.

## Place cells in the CA1 subregion of the hippocampus displayed remapping and decreased field widths following contextual fear conditioning

The hippocampus plays a critical role in the encoding, consolidation, and recall of memories and is necessary for the expression of contextual fear memory (*Anagnostaras et al., 2001*; *Ji and Maren, 2007*; *Liu et al., 2012*; *Maren et al., 2013*; *Maren and Holt, 2000*; *Phillips and LeDoux, 1992*; *Ramirez et al., 2013*; *Wiltgen and Silva, 2007*). Context is thought to be represented in the hippocampus by place cells that fire at specific spatial locations (place fields) in an environment (*O'Keefe and Dostrovsky, 1971*). These place cells have been thought to help discriminate between feared and neutral contexts. Previous studies on freely moving animals have shown that fear conditioning can cause place cells to remap (*Blair et al., 2023*; *Kim et al., 2015*; *Kinsky et al., 2023*; *Mamad et al., 2019*; *Moita et al., 2004*; *Ormond et al., 2023*; *Schuette et al., 2020*; *Wang et al., 2012*; *Wang et al., 2015*; *Wu et al., 2017*). Remapping is defined by place cells shifting their firing fields, indicative of place cells adapting to incorporate new information into memory (*Colgin et al., 2008*; *Leutgeb et al., 2005*; *Muller and Kubie, 1987*). Additionally, a recent study (*Schuette et al., 2020*) that performed calcium imaging of hippocampal cells in freely moving mice found that fear conditioning narrows the width of place fields in the feared environment. This suggests that place cells encode the same environment on a finer scale following fear conditioning, potentially enhancing context discrimination and threat avoidance (*Schuette et al., 2020*).

We replicated these findings in our head-fixed version of CFC in VR. To investigate the effect of fear conditioning on place cell activity in our task, we expressed a calcium indicator, GCaMP6f (*Chen et al., 2013*), in hippocampal pyramidal cells and, using two-photon microscopy, imaged from the same hippocampal neurons across days during fear conditioning and fear memory recall in Paradigm 1 (*Figure 7A–B*, n=8 mice). We found that a subset of place cells formed before CFC remained stable, while others remapped or changed their preferred firing locations after CFC (see examples in *Figure 7C and D–G*). This is similar to previously reported observations (*Blair et al., 2023*; *Moita et al., 2004*; *Wang et al., 2012*; *Wang et al., 2015*).

We found remapping to occur in both the Familiar and the CFC VR to similar extents, as indicated by the lower correlation across sessions/days compared to the within-session correlation (*Figure 7F–G*).

**Table 6.** Linear mixed effect (LME) summary (estimate ±standard error, p-value) for average freezing in Paradigm 3 with fixed effects (recall days, baseline: before CFC, Day 0) and random effect (mouse).

| VR | Recall Day 1 | Recall Day 2 | Recall Day 3 |
|---|---|---|---|
| Control VR | 2.65±0.97, p=0.008 | 3.99±0.97, p=0.0001 | 2.28±0.97, p=0.02 |
| CFC VR | 10.30±1.41, p<0.001 | 8.81±1.41, p<0.001 | 4.28±1.41, p=0.004 |

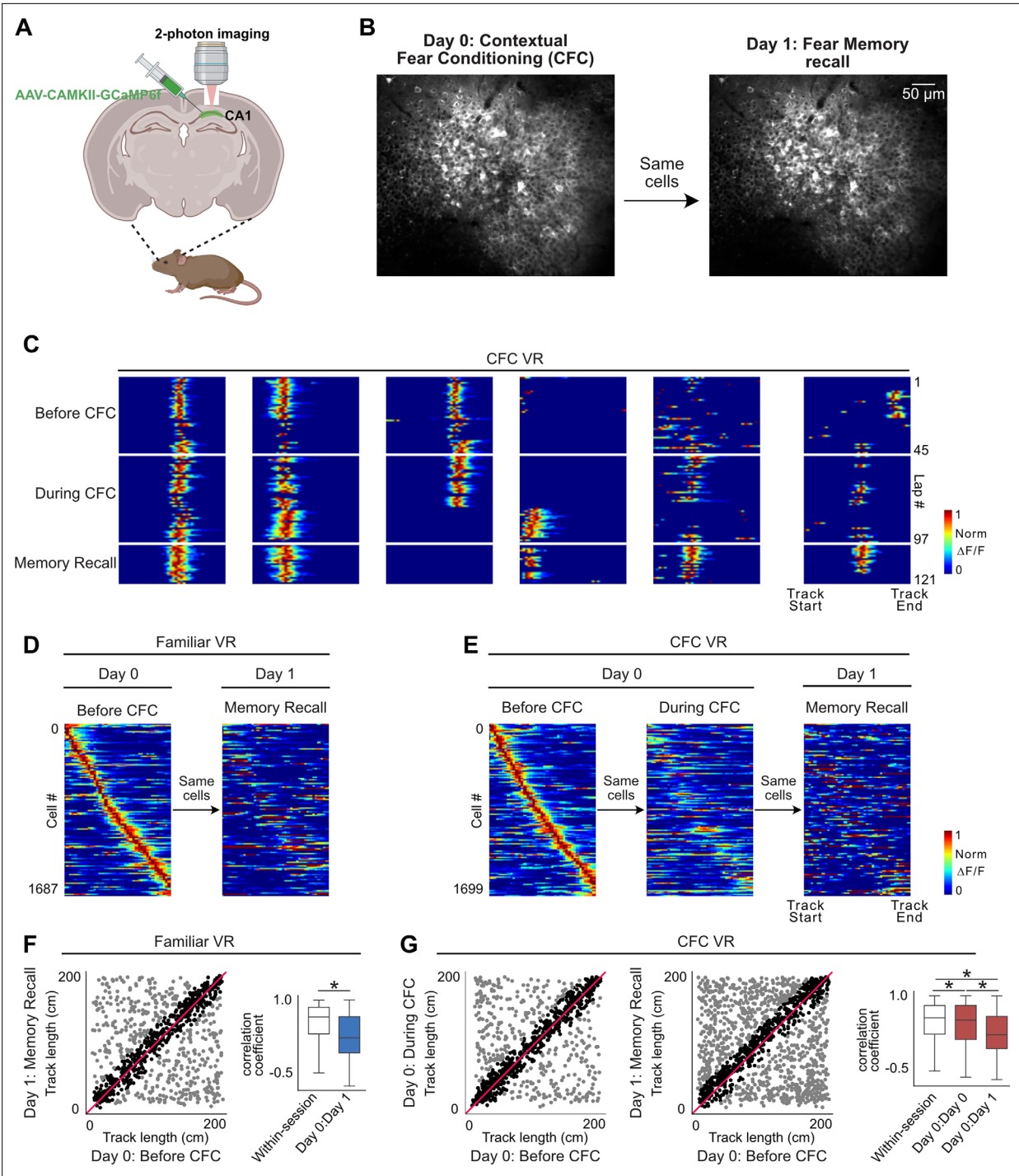

**Figure 7.** Fear conditioning results in place cell remapping in both Familiar and contextual fear conditioning (CFC) virtual reality (VR). (**A**) A schematic of the procedure for two-photon imaging of large populations of the same CA1 neurons over multiple days, created with BioRender.com (**B**) An example field of view across the two experiment days. The same field of view was aligned and imaged across days. (**C**) Examples of a few place cells in the CFC VR across sessions on Day 0 (before CFC, during CFC) and Day 1 (Memory Recall). White lines separate laps in each session. Some place cells maintain stable fields across days, while others remap. (**D**) Place fields defined on Day 0 in Familiar VR before CFC are plotted across Day 1 during fear memory recall (n=8 mice). (**E**) Place fields defined on Day 0 in CFC VR before CFC are plotted across the fear conditioning session and on Day 1 during fear memory recall. (**F**) On the left is a scatter plot of the center of mass of place fields defined in Familiar VR before CFC on Day 0 (x-axis) compared to their center of mass on Day 1 during fear memory recall (y-axis). On the right is a boxplot of correlation coefficients between mean place fields defined in Familiar VR before CFC on Day 0 and memory recall on Day 1 (Day 0: Day 1). The within-session correlation coefficients serve as control and were calculated between mean place fields in the first half and second half of the Familiar VR before the CFC session. (**G**) A scatter plot of the center of mass

*Figure 7 continued on next page*

*Figure 7 continued*

of place fields defined in CFC VR before CFC on Day 0 (x-axis) compared to their center of mass on (right) Day 0 during CFC and (middle) Day 1 during memory recall. On the left is a distribution of correlation coefficients between mean place fields defined in Familiar VR before CFC on Day 0 and during CFC (Day 0: Day 0) and memory recall on Day 1 (Day 0: Day 1). The within-session correlation was calculated between mean place fields in the first half and second half of the CFC VR before the fear conditioning session. Asterisk (*) indicates significant p-values (KS test, p<0.01). Our findings show that place fields present in the before CFC sessions in both Familiar and CFC VR showed significant remapping following fear conditioning.

This contradicts previous findings by *Moita et al., 2004*, who reported more stability in the control environment compared to the CFC environment after fear conditioning. This discrepancy may be explained by the different reward protocols used. Moita et al.'s rats foraged continuously for food pellets during the experimental sessions, while our mice received no rewards. We removed water rewards from the previously rewarded Familiar VR, and as we have shown in earlier work, removing rewards from a previously rewarded environment causes the place map to become more unstable and likely to remap across days (*Krishnan et al., 2022*; *Krishnan and Sheffield, 2023*). Additionally, Moita et al. measured place cell stability within a single day, with just 1 hr between conditioning and retrieval sessions, while our recordings were separated by a day, including intervening sleep periods. This longer interval may have contributed to more pronounced remapping compared to the shorter interval used by Moita et al (*Bollmann et al., 2025*; *Cai et al., 2009*; *Maboudi et al., 2018*).

Next, we looked at the place cells in each session. Interestingly, we found an increase in the number of place cells in the CFC VR during memory recall, a finding not previously reported (*Figure 8A–B*). We also found that place cells had narrower fields in the CFC VR during memory recall (*Figure 8C*), as previously shown (*Schuette et al., 2020*). We did not find a significant difference in other place cell

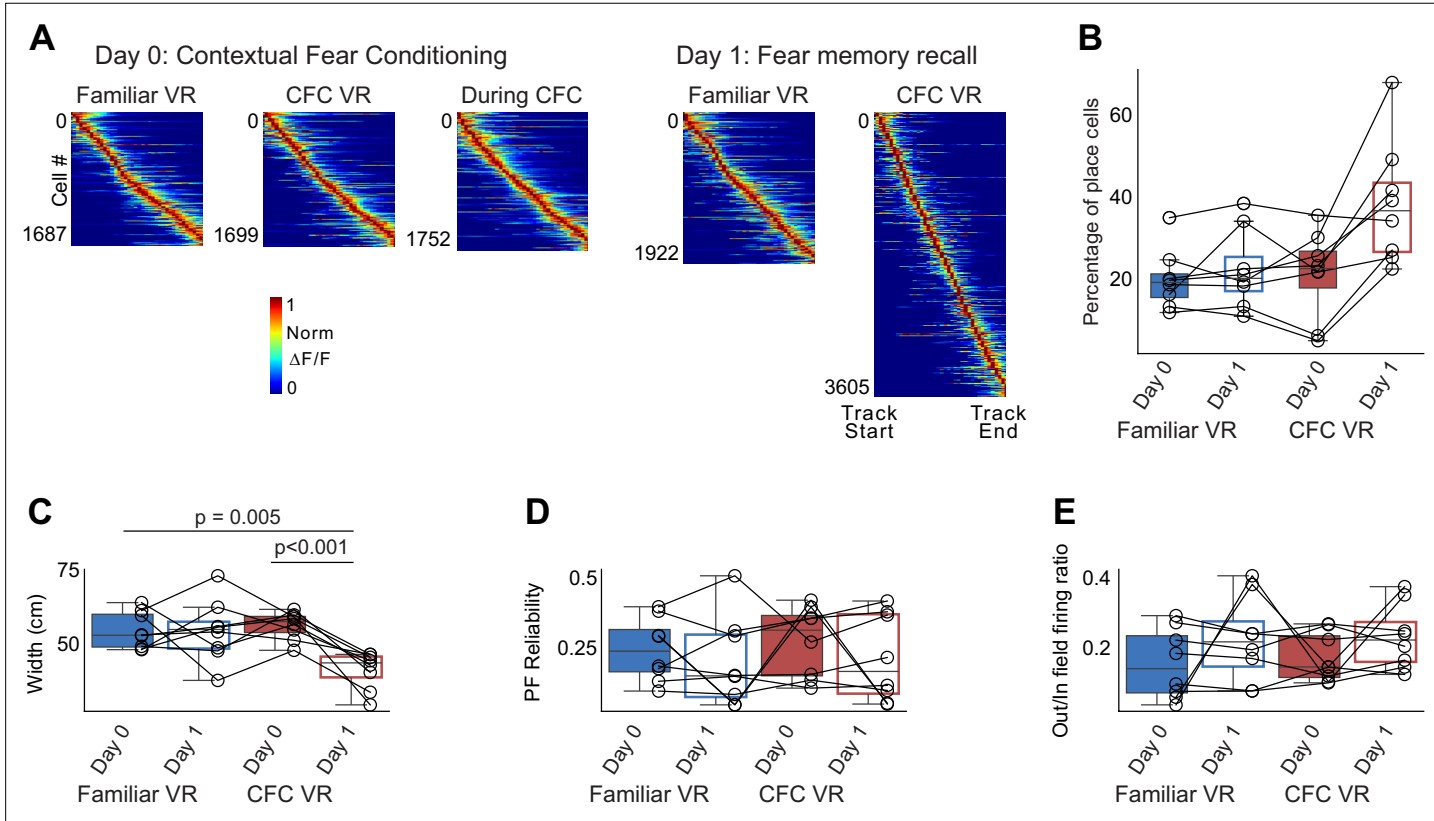

**Figure 8.** The widths of place fields in CA1 narrow during memory recall. (**A**) Place fields were defined and sorted by track length in each session, pooled from all mice (n = 8 mice). Each cell's activity was normalized to its peak, and cells were sorted by their center of mass along the track. (**B**) The percentage of place cells is calculated as the number of place cells divided by the total number of recorded cells. More place cells were identified by our algorithm in the CFC VR on Day 1. (**C–E**) Parameters of place fields: (**C**) Width, (**D**) Place Field (PF) Reliability, and (**E**) out/in field firing ratio in both VRs on Day 0 and Day 1. The width of the place fields in the CFC VR significantly decreased on Day 1. No other parameters significantly varied across days. p-values were calculated using a paired t-test.

parameters we examined, such as lap-by-lap reliability (*Figure 8D*) and out-of-field firing (*Figure 8E*). Thus, our observations of remapping and narrower place field widths replicate findings from freely moving animals, validating our paradigm's use to explore further questions using the head-fixed preparation to study the neural underpinnings of contextual fear memories. We add to this field by showing an increase in the proportion of place cells following CFC, a possible neural substrate of fear memory recall.

## Discussion

We present three paradigms for performing CFC in head-fixed mice, using a conductive 'tail-coat' to deliver aversive shocks as mice navigate VR environments on a treadmill. Our key findings are: (1) Head-fixed mice exhibited freezing behavior in response to an aversive stimulus in VR, consistent with conditioned responses seen in freely moving animals. Freezing was especially pronounced during the first lap in the fear-conditioned environment compared to the neutral environment across all paradigms tested. (2) With mice on a cylindrical treadmill, additional conditioned responses such as hesitation and backward movement were detected. (3) The choice of neutral environment influenced fear discrimination: mice froze more in the fear-conditioned VR when the neutral environment was familiar rather than novel. Freezing in the fear-conditioned VR could be enhanced by shorter ISIs and the presence of the tail-coat during recall when the neutral environment was novel. (4) Following fear conditioning, place cells in the hippocampal CA1 region showed remapping, increased numbers, and narrower fields, paralleling findings in freely moving animals and offering new insights.

Here, we experimented with three paradigms for fear conditioning in mice, with the main difference being the neutral environment where the mice didn't receive any shock. In Paradigm 1, the neutral VR was familiar, where the mice had previously received rewards, and the CFC VR was novel. In Paradigms 2 and 3, both the neutral and CFC VRs were similarly novel. In Paradigm 3, we decreased the ISI and kept the tail-coat on the animal when testing for memory recall. Thus, we have tested three different variations for performing CFC in head-fixed mice. On average, mice showed more freezing behavior in the CFC VR compared to the neutral VR. A common feature across all paradigms was pronounced freezing during the very first lap in the CFC VR. While mice in Paradigm 2 extinguished fear rapidly after the first lap, freezing behavior persisted up to 3–4 recall days in Paradigms 1 and 3, to varying degrees. We also observed variability in conditioned responses across mice. Some displayed an opposite pattern, freezing more in the neutral VR or equally in both VRs. This variability in behavior, similar to that observed in freely moving animals (*Chu et al., 2024*; *Dos Santos Corrêa et al., 2019*; *Navarro-Sánchez et al., 2024*; *Poulos et al., 2016*; *Russo and Parsons, 2021*; *Totty et al., 2021*), can be insightful when combined with investigations of neural activity, helping to understand the neural dynamics that contribute to an animal's ability to learn the association between a context and fearful stimuli. This can reveal the mechanisms by which such memories may generalize or become maladaptive, as seen in post-traumatic stress disorders.

We have shown here an example of using Paradigm 1 to study place cells in the hippocampus and their role in fear memory formation and recall. We longitudinally recorded from a large group of the same hippocampal CA1 cells and demonstrated that place cells remap and narrow their place field widths with fear conditioning. This occurs in head-fixed mice navigating VR environments, similar to freely moving animals. Narrow place fields may suggest sharper spatial tuning to incorporate a salient emotional memory and may aid in context discrimination (*Schuette et al., 2020*). However, how place cells narrow their widths in response to fear conditioning remains an open question. One possible way to answer this is by imaging CA1 place cell dendrites to understand how they integrate inputs before and after fear conditioning. We can explore what dendritic firing and plasticity mechanisms lead to narrow place field widths. We can then perturb these firing patterns to attribute causality to the narrowing of place field widths in fear discrimination. Our head-fixed paradigm enables the recording of dendrites and the study of plasticity mechanisms (*Sheffield et al., 2017*; *Sheffield and Dombeck, 2015*), thus facilitating these investigations.

We recently used Paradigm 3 to understand the role of the thalamic nucleus reuniens-hippocampal pathway (*Ratigan et al., 2023*). Using a two-photon microscope, we identified neural activity in this pathway by imaging axons from the nucleus reuniens that project to the CA1 with single-axon resolution. We found this pathway necessary for suppressing fear memory retrieval, context discrimination, and fear extinction. The robust freezing response in Paradigm 3 in the fear-conditioned VR,

compared to a neutral novel VR, allowed us to align our findings with the existing literature on the role of the nucleus reuniens-hippocampal circuit in fear memory in freely moving animals (*Ramanathan and Maren, 2019*; *Ramanathan et al., 2018*; *Totty et al., 2023*; *Troyner et al., 2018*; *Vetere et al., 2017*; *Xu and Südhof, 2013*). Furthermore, the subcellular resolution achieved through two-photon calcium imaging using the head-fixed preparation allowed us to longitudinally observe and understand the inputs that axons from the reuniens send to the hippocampus during fear memory formation and retrieval, a level of detail that is difficult to achieve with current techniques in freely moving preparations.

In another unpublished study, we used Paradigm 2 and identified co-active neurons between CA1 and CA3 subregions of the hippocampus during fear memory recall. The difference between the first and subsequent laps has helped us understand the progression of neural activity in the hippocampus as mice go from retrieving the fear memory to extinction within one session. Moreover, the variability in response across mice has allowed us to regress the neural activity to the behavior, identifying neural differences between those mice that accurately recall the fear memory and those that do not. These studies showcase the broad applicability of our different CFC paradigms.

In the laboratory, CFC provokes freezing behavior in freely moving rodents, as demonstrated by several studies that span decades (*Fanselow, 1980*; *Fanselow, 1990*; *Fendt and Fanselow, 1999*; *Maren, 2001*; *Maren et al., 2013*; *Phillips and LeDoux, 1992*; *Tovote et al., 2015*). In our study, we found that a mild electric shock applied to the tail could also trigger freezing behavior as the conditioned response in head-fixed mice, making our paradigm more comparable to the freely moving paradigms than previously attempted methods using air-puff stimulation (*Lovett-Barron et al., 2014*; *Rajasethupathy et al., 2015*). There are numerous opportunities for future work to systematically test how different parameters—such as shock amplitude, number and spacing of shocks, duration of fear conditioning, context pre-exposure time, the configuration of Control versus CFC VR environments, and the inclusion of additional sensory cues—may influence both behavioral outcomes during recall and extinction, as well as the underlying neural representations (*Fanselow, 1990*; *Huckleberry et al., 2016*; *Ji and Maren, 2007*; *Maren and Holmes, 2016*; *Poulos et al., 2016*). Our current study provides only an initial exploration of this parameter space.

In this manuscript, we focused on freezing as the primary conditioned response to align with studies of freely moving animals. However, we also see other defensive responses unique to our treadmill-based task, as well as fear-related hesitation and backward movements during memory recall—defensive behaviors that have also been observed in freely moving animals (*Chu et al., 2024*; *Furuyama et al., 2023*; *Totty et al., 2021*; *Trott et al., 2022*). Our technique enables us to conduct a more detailed analysis of the animal's behavior. The head-fixed setup allows us to easily measure physiological responses including pupil diameter, respiration, vocalization, heart rate, and body temperature. These parameters are known to change with fear conditioning (*Dupin et al., 2020*; *Godsil et al., 2000*; *Korn et al., 2017*; *Shionoya et al., 2013*; *Stiedl and Spiess, 1997*). Such physiological measurements can also help differentiate between the various conditioned responses we observe. There are also opportunities to use machine-learning-empowered algorithms to parse out new behavior markers for contextual fear memory, such as gestures or facial expressions (*Datta et al., 2019*; *Mathis and Mathis, 2020*; *Syeda et al., 2024*; *Wiltschko et al., 2020*). Thus, we can understand the animal's behavior in a well-controlled environment, which when combined with neural activity recording, can lead to new insights into the neural circuits mediating different defensive responses and emotional states.

Significant insights into memory mechanisms through the CFC task have also come from recent discoveries of memory engrams in the brain (*Josselyn et al., 2015*; *Josselyn and Tonegawa, 2020*; *Liu et al., 2012*; *Tonegawa et al., 2018*). Engram cells, identified in various brain regions with the help of genetic tagging and optogenetics, play a crucial role in encoding, consolidating, and retrieving specific memories. Typically, these cells are identified based on the expression of the immediate early gene *Fos* during fear conditioning. Their artificial activation with optogenetics can trigger memory recall, evidenced by increased freezing behavior. Understanding which neurons form these engram cells, how *Fos* expression is driven during memory formation, how memory information is stored in these cells, how they evolve with learning, and how their activation results in memory recall is an active area of research (*Frankland et al., 2024*; *Mocle et al., 2024*; *Monasterio et al., 2024*; *Pettit et al., 2022*; *Suthard et al., 2024*; *Uytiepo et al., 2024*). Incorporating our paradigm with

simultaneous measurements of cellular and subcellular activity can help answer some of these questions. For instance, a recent study (*Monasterio et al., 2024*) tagged engram cells during fear conditioning while animals were freely moving and then imaged the spontaneous dynamics of these cells using two-photon imaging before and after fear conditioning. The study revealed that cells forming engrams were inherently more active. A limitation of this study was the separate contexts for imaging versus tagging, which can be addressed with our paradigm. Our CFC paradigm allows for direct neural recording of these cells before, during, and after tagging in the same context to examine how activity during memory formation relates to engram cell emergence, population dynamics, and memory recall. Furthermore, holographic stimulation techniques can reactivate specific cell ensembles of interest to help understand the causal relationships between engram cells and memory. Moreover, the hippocampus contains both place cells and engram cells, both of which appear to be involved in contextual memory. However, their relationship is not yet fully understood. Our CFC paradigm offers an opportunity to study both place cells, as we have shown here, and engram cells simultaneously. This can be achieved by incorporating techniques that allow for in vivo labeling of *Fos* (*Pettit et al., 2022*). Thus, our head-fixed version of the CFC paradigm opens new avenues for research into the neural underpinnings of fear learning and memory.

Another advantage of our method is that it allows for studying changes in neuronal dynamics in response to an aversive stimulus, such as a shock. Our fields of view during the shock and subsequent flight-like response periods remained stable, and we could correct image movements during these periods with Suite2p (*Pachitariu et al., 2017*). This is a significant improvement over one-photon and electrophysiological techniques, where abrupt movements such as those induced by an electric shock can change fields of view and reduce the likelihood of recording from the same groups of cells, thereby hindering longitudinal assessments of neural dynamics. In Paradigm 1, where we imaged hippocampal cells during shock, we did not find cells responding to the shock itself, as others have reported (*Barth et al., 2023*; *Mocle et al., 2024*; *Suthard et al., 2024*). We suspect this could be due to the number or duration of shocks administered and our shocks not being associated with a specific cue or location. In future studies, we plan to optimize these parameters to understand better how neural dynamics change in response to an aversive cue.

Finally, we recognize that our head-fixed CFC paradigm has some limitations. Compared to the freely moving CFC, our setup is less naturalistic. We defined different contexts as two visually distinct VR environments while odor, sound, and tactile cues remained constant. This could have made it harder for mice to differentiate between the two contexts, particularly in Paradigm 2. Unlike freely moving animals in CFC studies, our mice were water-restricted throughout the experimental process. Because the animals need to get used to wearing a tailcoat and running in the VRs without a reward, we tend to 'over-train' the mice. This criterion excludes mice that are less adept at running, limiting the number that go through the process and requiring larger cohorts of mice. This also presents scalability challenges compared to freely moving paradigms, as successful implementation of the head-fixed CFC approach requires weeks of reward-based training, larger sample sizes, and a technically demanding VR setup that integrates multiple hardware and software components to precisely monitor behavior. Additionally, while rewards were removed during experimental sessions, water rewards were still used to motivate running during training. This reward-associated training could subtly influence later behavior. For instance, the increased freezing in the Familiar VR on later recall days and remapping of place cells in that context could reflect the lingering effects of reward-associated training. While Paradigms 2 and 3 address many of these concerns, our approach still differs from freely moving paradigms due to this initial reward-based training. The impact of using rewards during training on recall behavior needs further investigation. Furthermore, despite our best efforts to include equal numbers of male and female mice, only some female mice reached the experimental stage. This could be due to the tail-coat weight and the reduced weight of females after water restriction compared to males. Female mice of the same age (10–12 wk) weighed significantly less than males (20–23 g vs 25–30 g) and lost further weight following water restriction. Although we observed no behavioral differences between the four tested females and the male mice, we cannot guarantee similar results in other female animals without further testing. Studies indicate that different sexes may display varying conditioned responses, as seen in rats (*Gruene et al., 2015*; *Maren et al., 1994*). In the future, we aim to address these limitations.

In conclusion, we have developed a head-fixed version of the classic CFC paradigm. Our findings demonstrate that mice exhibit freezing behavior when fear-conditioned in a VR environment. Moreover, we observed that hippocampal place cells remap and narrow their fields in response to fear conditioning, mirroring observations in freely moving animals. This innovative paradigm bridges the behavioral task gap for techniques requiring head-fixation, paving the way for new investigations into neural circuits and deepening our understanding of the neural foundations of emotional memory.

## Methods

### Subjects

All experimental and surgical procedures adhered to the University of Chicago Animal Care and Use Committee guidelines under protocol number: 72508. This study utilized 10–12 wk-old C57BL/6 J (Strain #:000664), Thy1GCaMP6s (Strain #:024275), and Tg(Grik4-cre)G32-4Stl/J (Strain #:006474) male and female mice obtained from JAX labs. Following the start of water restriction, the mice were individually housed in a reverse 12 hr light/dark cycle, and behavioral experiments were conducted during the animals' dark cycle.

### Head-fixation surgery

Mice were anesthetized with 1–2% isoflurane and given an intraperitoneal injection of 0.5 mL of saline and a subcutaneous injection of 0.45 mL of Meloxicam (1–2 mg/kg). A small incision was made on the skull to attach a metal head-plate (9.1 mm × 31.7 mm, Atlas Tool and Die Works) to head-restrain the mice. Some mice also had an imaging window implanted above the hippocampus, as previously described (*Dong et al., 2021*; *Krishnan et al., 2022*; *Sheffield and Dombeck, 2015*). Post-surgery mice were housed individually. After a recovery day, they were water-restricted to 0.8–1.0 ml/day.

### Behavior training

Head-restrained mice were trained to run on a cylindrical treadmill to navigate VR environments similar to the setups previously described (*Dong et al., 2021*; *Krishnan et al., 2022*; *Sheffield and Dombeck, 2015*). Behavioral training began 7 d following the water restriction. Briefly, VR environments were created using the Virtual Reality MATLAB Engine, or VIRMEN (*Aronov and Tank, 2014*), and projected to five screens covering the mouse's field of view. Mice were head restrained with their limbs resting on a freely rotating Styrofoam wheel, which acted as a treadmill. VR environments were 2 m linear tracks. Mice received a water reward (4 μl) through a waterspout upon completing each traversal of the linear track, considered a lap. This reward was followed by a 1.5 s pause in the VR to allow the mouse to consume the water. The mice were then virtually teleported to the start of the track to begin a new lap. Training sessions lasted 30 min per day. Mice were considered well-trained when they could run at >3 laps per minute. We found that this level of training was necessary to ensure that mice would continue to run even without rewards (*Krishnan et al., 2022*; *Krishnan and Sheffield,*

**Table 7.** Items required for head-fixed contextual fear discrimination.

| Item | Manufacturer | Link |
|---|---|---|
| Denim patch cloth | Any fabric or hobby shop | https://www.michaels.com/product/loops-threads-denim-patches-assorted-10113795 |
| Snap Buttons | Any fabric or hobby shop | https://www.michaels.com/product/loops-threads-sew-on-snaps-10354228?michaelsStore=1191&inv=14 |
| Conductive nylon fabric tape | Adafruit | https://www.adafruit.com/product/3960 |
| Stainless conductive thread – 3 Ply | Adafruit | https://www.adafruit.com/product/641 |
| Multipurpose 304 Stainless steel sheet | McMaster-Carr | https://www.mcmaster.com/products/304-stainless-steel/multipurpose-304-stainless-steel-6/?s=304-stainless-steel |
| Shock box | Colbourn precision animal shocker Alternate: Lafayette Instruments Scrambled Grid Current Generator | https://lafayetteinstrument.com |

2023). Generally, about 60% of mice met our criteria, which typically took around 10–14 d to achieve. Behavior data was collected using a PicoScope Oscilloscope (PICO4824, Pico Technology, v6.13.2).

## Tail-coat

A custom-designed 'tail-coat' delivered mild electric shocks to a head-fixed mouse's tail during VR navigation. A pictorial representation of the tail-coat can be found in *Figure 1D*. The tail-coat, hand-made from wearable conductive fabrics, featured a lightweight design to ensure the mouse could continue running during baseline VR exploration (*Table 7*). First, the mouse's tail was measured while anesthetized for head-plate surgeries. A small denim patch cloth (Michael's) was cut into a rectangle to fit snugly on the mouse's tail. Marks were made where snap buttons (Michael's) would be sewn, ensuring they clipped just above the tail. Given that the mouse's tail narrows away from the body, the buttons closer to the body were placed slightly further apart than those closer to the tip. Typically, the buttons were set 2 cm apart on the narrow end and 2.5 cm apart on the broader end. Care was taken to adjust the placement of the buttons so they weren't too tight or loose for the mouse. We found that these measurements were generally similar across sexes and for the age group that we tested. After measurements were taken and the cloth cut, small strips of 8 mm wide conductive fabric tape (Adafruit) were attached to either side of the cloth, with some tape hanging off the edge. A metal strip (McMaster Carr) of the same width was affixed to this portion of the tape to provide a secure grip for alligator clips used in fear conditioning. Finally, snap buttons were sewn with conductive thread at the marked spots. Clear nail polish was applied to secure the threads in place. The tail-coat creation process should take less than half an hour. Dimensions of the cloth for a typical male mouse, 12 wk of age and weighing ~30 g, are shown in *Figure 1D*. The denim cloth was 4 cm in width and 3.2 cm in length; the conductive tape was 1.2 cm in width and 5.2 cm in length, with 1.5 cm hanging out at each side.

## Contextual fear conditioning protocol in head-fixed mice

Once the mice were well-trained and met our criteria of running >3 laps per minute, they were ready for the next stage. The following day, a tail-coat was attached to the mouse's tail. Mice that maintained consistent running in the VR, with minimal pauses, while wearing the tail-coat advanced to the next stage. For mice startled by the tail-coat, we typically attempted another day with the tail-coat, rewarding them and observing if their running behavior improved to running at least three laps per minute. If not, these mice were not advanced to the next stage. In mice that do advance, on the subsequent day, the tail-coat was kept on, and the water reward was removed. All else remained the same, including the 1.5 s pause at the end, to allow animals to distinguish the start and end of a lap. If the mice continued to run as before, they moved on to the experiment stage. From that day forward, the mice no longer received water reinforcement when running in the VR environments. Typically, about 40% of mice made it to the experiment stage. Tail-coats were only added during VR exploration and removed before returning the animals to their home cage. Once the mice advanced to the experiment stage, they were taken through the entire process and not excluded from analysis, even if they showed signs of inconsistent running behavior, which happened with some mice (for instance, see *Figure 3—figure supplement 1B–C*).

We implemented three distinct paradigms to assess contextual fear discrimination. In the first paradigm, we conducted experiments over 2 d (Paradigm 1). On Day 0, mice engaged with the same VR in which they were trained, now referred to as the Familiar VR. After 10 min, we transitioned them to a new VR for another 10 min; this new VR would later be associated with shocks (CFC VR). During this phase, the mice explored both VRs without any rewards while wearing the tail-coat. After the initial exploration in the CFC VR, they received six mild electric shocks, each lasting 1 s with 1 min intervals in between. The shocks were administered using a shock generator (Colbourn Precision animal shocker, Harvard Bioscience). A TTL pulse was generated using custom-written codes in MATLAB to turn on the shock generator via an Arduino. The shock times were recorded by the PicoScope Oscilloscope (PICO4824, Pico Technology, v6.13.2). We tested different shock intensities ranging from 0.5 to 1.2 mA. After the last shock, mice were removed from the VR and returned to their home cage. The following day (Day 1), we placed the mice in the Familiar VR or CFC VR for 10 min to test their memory recall. No rewards were delivered during this phase; however, the tail-coat was not added during memory recall. Some mice underwent additional recall tests over subsequent days to assess

the time required for fear memory extinction. As a control, some mice underwent the same paradigm but without any shocks.

In the second paradigm (Paradigm 2), the experiment spanned over 3 d. On Day - 1, mice explored two novel VRs for 10 min each, establishing a baseline for context exploration. One of these VRs served as a control, while the other, the CFC VR, was where the mice would later receive electric shocks. On Day 0, the mice again spent 10 min each in the Control and CFC VRs. After this, they received six mild electric shocks, each lasting 1 s, with a strength of 1 mA and an inter-trial interval of 1 min. The mice wore the tail-coat on both Days –1 and 0. On Day 1, the mice were placed in either the Control VR or CFC VR for 10 min each, in a counterbalanced manner, to test for memory recall. No rewards were given on all the experimental days; however, the tail-coat was not added to the tails on the recall day (Day 1). The third paradigm (Paradigm 3) was similar to Paradigm 2 and used two novel VRs, except that a tail-coat was also added to the mice's tails during recall tests.

Additionally, a subset of mice in Paradigms 1 and 2 and all mice in Paradigm 3 went through multiple days of recall tests. The subset of mice was randomly chosen, not based on their recall behavior from the previous day. Mice were presented with either the CFC VR or the Control VR in a counterbalanced manner.

All VRs were randomly assigned as either Control or CFC VR to prevent biasing any inherent preferences in mice. However, the Familiar VR was always kept the same. We also used both sexes in our experiments. However, we found female mice less receptive to the 'tail-coat' than their male counterparts. We believe this is due to the low average weight of female mice (~16 g) following water restriction. Female mice of the same age (10–12 wk) weighed significantly less than males (20–23 g vs 25–30 g) and lost further weight following water restriction. As a result, only four female mice could proceed through and complete all the behavioral tests.

## Behavior measurement

Freezing periods were periods during which the animal's instantaneous velocity was 0 cm/s. Freezing epochs were defined as continuous, uninterrupted freezing periods longer than 1 s. The time taken to complete a lap was calculated as the total time (in seconds) the animal took to run from 0 to 200 cm.

## Surgery and two-photon imaging

Stereotaxic surgeries and cannula window implantation above the CA1 subregion of the hippocampus were performed in the same way as previously reported (*Krishnan et al., 2022*). Briefly, a genetically-encoded calcium indicator, AAV1-CamKII-GCaMP6f (pENN.AAV.CamKII.GCaMP6f.WPRE.SV40 was a gift from James M. Wilson – Addgene viral prep #100834-AAV1; https://www.addgene.org/100834/; RRID:Addgene_100834) was injected (~50 nL at a depth of 1.25 mm below the surface of the dura) using a beveled glass micropipette leading to GCaMP6f expression in a large population of CA1 pyramidal cells. Mice underwent water restriction (1.0 ml/day) following viral injection and, 7 d later, were implanted with a hippocampal window and head plate (*Dombeck et al., 2010*). Mice were allowed to rest from the surgeries for 3–4 d, after which behavioral training began. Imaging was done using a laser scanning two-photon microscope (Neurolabware). The setup is the same as the one previously described (*Krishnan et al., 2022*). Time-series images were collected through Scanbox (v4.1, Neurolabware), and the PicoScope Oscilloscope (PICO4824, Pico Technology, v6.13.2) was used to synchronize frame acquisition timing with behavior. At the end of the imaging session on Day –1 or Day 0, a 1 min time-series movie was collected at a higher magnification and then averaged to aid as a reference frame in finding the same imaging plane on subsequent days. Across-day images collected were motion-corrected using Suite2p (*Pachitariu et al., 2017*) and aligned using ImageJ (v1.53, NIH). Region of interests were extracted using Suite2p.

## Place cell extraction and place field parameters

Place fields were identified as previously described (*Dong et al., 2021*; *Krishnan et al., 2022*). Briefly, the 2 m track was divided into 40 position bins (each 5 cm wide). The running behavior of the animal was filtered to exclude periods where the animal was immobile (speed <1 cm/s). Filtering ensured that place cells were defined only during active exploration and not during freezing bouts. We have previously found that filtering versus not filtering yields similar results (*Krishnan et al., 2022*). Extracted place fields satisfied the following criteria, which were used for all conditions and all mice: 1. The

average ΔF/F was greater than 10% above the baseline. 2. The cell displayed significant calcium transients on >30% of laps in the field. 3. The rising phase of the mean transient was located on the track. 4. Their p-value from bootstrapping was <0.05. Multiple place fields within the same cell were treated independently. Parameters of place fields were calculated as described before (*Krishnan et al., 2022*); this includes the center of mass (COM), place field reliability, out/in-field firing ratio, and place field widths.

## Statistics

At the time of study design, sample sizes were based on prior studies using head-fixed mice in virtual reality environments (*Krishnan et al., 2022*; *Sheffield and Dombeck, 2015*). However, the final sample size was constrained by the low attrition rate of mice progressing to the experimental stage. Mice were randomly assigned to one of three experimental setups and independently to a shock paradigm. Behavioral analyses, including the identification of freezing epochs, were conducted blind to the assigned paradigm until all data were compiled for group-level comparisons. The experiment has been independently replicated multiple times in the laboratory across different cohorts of mice and by different experimenters.

For related samples, we performed a paired t-test. Multiple comparisons of related samples were corrected with Bonferroni post-hoc. To quantify freezing across days, we employed a linear mixed-effects model with days of recall as the fixed effect and behavior prior to fear conditioning on Day 0 as baseline. The model also included a random effect for mice to account for the nesting of observations within individual subjects. Separate models were used for each VR. Boxplots are plotted to display the entire distribution of the data. The box in the boxplot ranges from the first quartile (25th percentile) to the third quartile (75th percentile), and the box shows the interquartile range (IQR). The line across the box represents the median (50th percentile). The whiskers extend to 1.5*IQR on either side of the box, and anything above this range is defined as an outlier. A two-tailed Kolmogorov-Smirnov (KS) Test was used to compare distributions. When not using boxplots, mean and confidence intervals are displayed.

## Acknowledgements

We thank the members of the Sheffield Lab for their comments on the manuscript, helpful discussions, and feedback throughout the process. This work was supported by The Whitehall Foundation, The Searle Scholars Program, The Sloan Foundation, The University of Chicago Institute for Neuroscience start-up funds and the National Institute for Health (1DP2NS111657-01, 1RF1NS127123-01) awarded to MS. and a T32 training grant (T32DA043469) and K01 (K01DA060994-01) from the National Institute on Drug Abuse awarded to SK.

## Additional information

### Funding

| Funder | Grant reference number | Author |
| --- | --- | --- |
| National Institute on Drug Abuse | T32DA043469 | Seetha Krishnan |
| National Institute on Drug Abuse | K01DA060994 | Seetha Krishnan |
| BRAIN Initiative | 1RF1NS127123 | Mark Sheffield |
| National Institute of Neurological Disorders and Stroke | 1DP2NS111657 | Mark Sheffield |
| Whitehall Foundation | | Mark Sheffield |
| Searle Scholars Program | | Mark Sheffield |
| Alfred P. Sloan Foundation | | Mark Sheffield |

| Funder | Grant reference number | Author |
|---|---|---|
| The University of Chicago Institute for Neuroscience startup funds | | Mark Sheffield |

The funders had no role in study design, data collection and interpretation, or the decision to submit the work for publication.

## Author contributions

Seetha Krishnan, Conceptualization, Data curation, Software, Formal analysis, Supervision, Validation, Investigation, Visualization, Methodology, Writing – original draft, Project administration, Writing – review and editing; Can Dong, Data curation, Software, Formal analysis, Validation, Investigation, Visualization, Methodology, Writing – review and editing; Heather Ratigan, Data curation, Validation, Investigation, Writing – review and editing; Denisse Morales-Rodriguez, Data curation, Investigation, Writing – review and editing; Chery Cherian, Data curation, Writing – review and editing; Mark Sheffield, Conceptualization, Resources, Supervision, Funding acquisition, Project administration, Writing – review and editing

## Author ORCIDs

Seetha Krishnan ⓘ https://orcid.org/0000-0002-6218-3995
Can Dong ⓘ https://orcid.org/0000-0002-4368-4858
Denisse Morales-Rodriguez ⓘ https://orcid.org/0000-0002-3272-7023
Mark Sheffield ⓘ https://orcid.org/0000-0003-0969-7820

## Ethics

All experimental and surgical procedures were approved by the University of Chicago Animal Care and Use Committee under protocol number: 72508 and carried out in strict accordance with their guidelines. All surgery was performed under isoflurane anesthesia, and every effort was made to minimize suffering.

Reviewer #1 (Public review): https://doi.org/10.7554/eLife.105422.3.sa1
Reviewer #2 (Public review): https://doi.org/10.7554/eLife.105422.3.sa2
Reviewer #3 (Public review): https://doi.org/10.7554/eLife.105422.3.sa3
Author response https://doi.org/10.7554/eLife.105422.3.sa4

# Additional files

## Supplementary files
MDAR checklist

## Data availability

Behavior data and processed imaging data can be found on Dryad at: https://doi.org/10.5061/dryad.m63xsj4f3. Scripts used for data analysis are available on GitHub at https://github.com/seethakris/CFCMethodspaper (copy archived at *Krishnan, 2025*).

The following dataset was generated:

| Author(s) | Year | Dataset title | Dataset URL | Database and Identifier |
|---|---|---|---|---|
| Seetha Lakshmi K, Can D, Heather R, Denisse MR, Chery C, Mark S | 2025 | A contextual fear conditioning paradigm in head-fixed mice exploring virtual reality [Dataset] | https://doi.org/10.5061/dryad.m63xsj4f3 | Dryad Digital Repository, 10.5061/dryad.m63xsj4f3 |

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
