## [Editor Report · eLife Assessment]

This **useful** study presents a virtual reality-based contextual fear conditioning paradigm for head-fixed mice. **Solid** evidence supports the claim that the reported methods provide a reliable paradigm for studying contextual fear conditioning in head-fixed mice. The approach provides a way to perform multiphoton imaging of neural circuits, and other techniques that are typically performed in head-fixed animals, during behaviors that have traditionally been studied in freely moving animals.

---

## [Referee Report · Reviewer #1 (Public review)]

The authors have developed a contextual fear learning (CFC) paradigm in head-fixed mice that produces freezing as the conditioned response. Typically, lick suppression is the conditioned response in such designs, but this (1) introduces a potential confounding influence of reward learning on neural assessments of aversion learning and (2) does not easily allow comparison of head-fixed studies with extensive previous work in freely moving animals, which use freezing as the primary conditioned response. This report describes 3 versions of this virtual reality CFC paradigm, its validation using place-cell remapping, and provides suggestions for further refinement and application.

The first part of this study is a report on the development and outcomes of 3 variations of the CFC paradigm in a virtual reality environment. The fundamental design is strong, with head-fixed mice required to run down a linear virtual track to obtain a water reward. Once trained, the water reward is no longer necessary and mice will navigate virtual reality environments. There are rigorous performance criteria to ensure that mice that make it to the experimental stage show very low levels of inactivity prior to fear conditioning. These criteria do result in only 40% of the mice making it to the experimental stage, but high rates of activity in the VR environment is crucial for detecting learning-related freezing. It is possible that further adjustments to the procedure could improve attrition rates.

Paradigm versions 1 and 2 vary the familiarity of the control context while paradigm versions 2 and 3 vary the inter-shock interval. Version 1 is the most promising, showing the greatest increase in conditioned freezing (~40%) and good discrimination between contexts (delta ~15-20%). Version 2 showed no clear evidence of learning - average freezing at recall day 1 was not different than pre-shock freezing. First lap freezing showed a difference, but this single lap effect is not useful for many of the neural circuit questions for which this paradigm is meant to facilitate. Version 3 produces greater freezing and slower extinction than version 2. While the magnitude of the context discrimination is less than that in version 1, further optimization of the VR CFC is likely to produce robust learning and extinction. The authors discuss several options for further optimization.

The second part of the study is a validation of the head-fixed CFC VR protocol through demonstration that fear conditioning leads to remapping of dorsal CA1 place fields, similar to that observed in freely moving subjects. The results support this aim and largely replicate previous findings in freely moving subjects. One difference from previous work of note is that VR CFC led to remapping of the control environment, not just the conditioning context. The authors present several possible explanations for this lack of specificity to the shock context. While this experiment examined place cell remapping after fear conditioning, it did not attempt to link neural activity to the learned association or freezing behavior.

In summary, this is an important methodological innovation and this study sets the initial parameters and neuronal validation needed to further optimize a head-fixed CFC paradigm that produces freezing. In the discussion, the authors note the limitations of this study, suggest next steps in refinement, and point to several future directions using this protocol to significantly advance our understanding of the neural circuits of threat-related learning and behavior.

Comments on revisions:

The manuscript is much stronger with the additions and revisions the authors provided in their revised submission.

---

## [Referee Report · Reviewer #2 (Public review)]

Summary:

In this manuscript, Krishnan et al devised three paradigms to perform contextual fear conditioning in head-fixed mice. Each of the paradigms relied on head-fixed mice running on a treadmill through virtual reality arenas. The authors tested the validity of three versions of the paradigms by using various parameters. The authors have addressed some of my initial concerns in their revised manuscript.

Strengths:

The authors have devised three new contextual fear conditioning paradigms in head-fixed mice. The authors tested a number of parameters towards optimization of this approach.

Weaknesses:

While some experimental parameters were tested in the manuscript, it appears that a large amount of additional testing and optimization will be required before reliable behavioral responses can be acquired and ultimately for the paradigm(s) to be useful for answering biological questions. One major factor will be optimizing parameters such that head-fixed mice in this paradigm can (largely) recapitulate what is observed in freely behaving mice. This may be challenging however, as they have previously published one of the three paradigms and the extensive additional testing they did in this current manuscript did not greatly improve the experimental setup. This may indicate limited immediate usefulness for the community as significant work likely remains for optimization.

Achievement of Aims:

The authors have put a significant amount of work in testing the paradigms, and as a result, progress has been made towards their usefulness in the field. However, a significant amount of optimization likely exists.

Impact on the field:

The development of a reliable paradigm for studying contextual fear in head-fixed animals would be a strong contribution to the field as it would enable sophisticated cell and circuit imaging analyses. This study is a good start towards this goal, but significant optimization is required for the paradigm(s) to fully benefit the field - especially to allow those who may have less experience in these approaches to use it in their own research.

---

## [Referee Report · Reviewer #3 (Public review)]

Summary:

Krishnan et al. present a novel contextual fear conditioning (CFC) paradigm using a virtual reality (VR) apparatus to evaluate whether conditioned context-induced freezing can be elicited in head-fixed mice. By combining this approach with two-photon imaging, the authors aim to provide high-resolution insights into the neural mechanisms underlying learning, memory, and fear. Their experiments demonstrate that head-fixed mice can discriminate between threat and non-threat contexts, exhibit fear-related behavior in VR, and show context-dependent variability during extinction. Supplemental analyses further explore alternative behaviors and the influence of experimental parameters, while hippocampal neuron remapping is tracked throughout the experiments, showcasing the paradigm's potential for studying memory formation and extinction processes.

Strengths:

Methodological Innovation: The integration of a VR-based CFC paradigm with real-time two-photon imaging offers a powerful, high-resolution tool for investigating the neural circuits underlying fear, learning, and memory.

Versatility and Utility: The paradigm provides a controlled and reproducible environment for studying contextual fear learning, addressing challenges associated with freely moving paradigms.

Potential for Broader Applications: By demonstrating hippocampal neuron remapping during fear learning and extinction, the study highlights the paradigm's utility for exploring memory dynamics, providing a strong foundation for future studies in behavioral neuroscience.

Comprehensive Data Presentation: The inclusion of supplemental figures and behavioral analyses (e.g., licking behaviors and variability in extinction) strengthens the manuscript by addressing additional dimensions of the experimental outcomes.

Weaknesses:

Optimization: many parameters remain to be tested in the VR fear conditioning paradigm.

Extended training and attrition rate: the paradigm requires weeks of training and only 40% of mice reach criteria.

---

## [Author Response]

The following is the authors’ response to the original reviews

We thank all the reviewers for their time and valuable feedback, which helped us improve our manuscript. Based on the comments, we have made several critical changes to the revised manuscript.

(1) We have changed our threshold for detecting freezing epochs from 1 cm/s to 0 cm/s in this revised manuscript. This change allows us to capture periods when animals are completely still on the treadmill, better matching the "true freezing" behavior seen in freely moving set-ups. We have added a new supplementary video (Supplementary Video 2) that better demonstrates the freezing response we observe. All results and figures in the revised manuscript reflect this updated threshold (Figure 2-6, Supplementary Figures 16, Tables 1-6). Our main findings remain robust, demonstrating that freezing serves as a reliable conditioned response in our paradigms, comparable to freely moving animals. Specifically, freezing behavior increased reliably in the fear-conditioned environment following CFC across all paradigms. We have also added data from a no-shock control group (Supplementary Figure 2) which, when compared to the conditioned group, shows that freezing responses in the conditioned group result from fear conditioning rather than immobility. We do observe other avoidance behaviors unique to our treadmill-based task— such as hesitation, backward movement, and slow crawls. These conditioned behaviors are captured through a separate metric: the time taken to complete a lap.

(2) As suggested by the reviewers, we have separately analyzed fear discrimination and extinction dynamics across recall days (Supplementary Figures 2, 5 and 6, Table 1-6). To assess fear discrimination, we use within-group comparisons to evaluate how well animals differentiate between the two VRs across days. For extinction, we use within-VR comparisons to examine freezing dynamics over time. Freezing across recall days is compared to baseline freezing (pre-conditioning) using a Linear Mixed Effects model (Tables 1-6), with recall days as fixed effects and mouse as a random effect, using baseline freezing as the reference.

(3) We have expanded the behavioral dataset in Paradigm 1 to investigate the effect of shock amplitude on the conditioned fear response (Supplementary Figure 2 C-E). Consistent with findings in freely moving animals, our data show that increasing shock intensity from 0.6 mA to 1.0 mA leads to stronger freezing. For the revised manuscript, we specifically increased the sample size in the 0.6 mA group (n = 8) in Paradigm 1, as this intensity is used in Paradigm 3. These additional data demonstrate that combining a lower shock amplitude with shorter inter-shock intervals and retaining the tail-coat during recall can enhance freezing, suggesting that these parameters help compensate for lower shock intensity.

(4) We have added more sample sizes to the imaging dataset (now n = 8, Figures 7-8).

Finally, we acknowledge that many aspects of this paradigm still require optimization. The headfixed CFC paradigm is in its early stages compared to the decades of research dedicated to understanding fear learning parameters in freely moving CFC paradigms. While there are numerous parameters that could be tested—both those identified through our own discussions and those raised by the reviewers—it is not feasible for a single lab to conduct a full evaluation of all the possible factors that could influence CFC in the head-fixed prep. A key limitation is that our approach requires robust navigation behavior in the VR without rewards, which requires weeks of training per mouse. It also necessitates larger sample sizes at the outset as not all animals will make it through our behavioral criteria required for CFC. Another important consideration is scalability. Unlike freely moving CFC paradigms, which allow parallel testing of many animals with minimal pre-training, the VR-CFC setup requires several weeks of behavior training and involves a more complex integration of hardware and software to accurately track behavior in virtual space. The number of VR rigs that can be operated simultaneously in a single lab is often limited, making high-throughput testing more challenging. These factors mean that the testing of a single parameter in a group of animals requires approximately 3–4 months to complete. Despite these constraints, we are committed to continue refining this paradigm over time. With this manuscript, our main aim was to provide a detailed framework, initial parameters, and evidence for conditioned behavior in the head-fixed preparation. By doing so, we hope to facilitate the adoption of this paradigm by researchers interested in studying the neural correlates of learning and memory using multiphoton imaging and stimulation techniques. This approach enables investigations that are not possible in freely moving animals, while the presence of freezing as a conditioned response allows for direct comparisons to the extensive body of work done in freely moving paradigms. Moving forward, we anticipate that optimizing this paradigm and identifying the key parameters that drive learning will be a collaborative, community-led effort.

**Public Reviews:**

**Reviewer #1 (Public review):**
The authors set out to develop a contextual fear learning (CFC) paradigm in head-fixed mice that would produce freezing as the conditioned response. Typically, lick suppression is the conditioned response in such designs, but this (1) introduces a potential confounding influence of reward learning on neural assessments of aversion learning and (2) does not easily allow comparison of head-fixed studies with extensive previous work in freely moving animals, which use freezing as the primary conditioned response.The first part of this study is a report on the development and outcomes of 3 variations of the CFC paradigm in a virtual reality environment. The fundamental design is strong, with headfixed mice required to run down a linear virtual track to obtain a water reward. Once trained, the water reward is no longer necessary and mice will navigate virtual reality environments. There are rigorous performance criteria to ensure that mice that make it to the experimental stage show very low levels of inactivity prior to fear conditioning. These criteria do result in only 40% of the mice making it to the experimental stage, but high rates of activity in the VR environment are crucial for detecting learning-related freezing. It is possible that further adjustments to the procedure could improve attrition rates.

We acknowledge that further adjustments to the procedure could improve attrition rates, and we will continue to work on improving the paradigm.

Paradigm versions 1 and 2 vary the familiarity of the control context while paradigm versions 2 and 3 vary the inter-shock interval. Paradigm version 1 is the most promising, showing the greatest increase in conditioned freezing (~40%) and good discrimination between contexts (delta ~15-20%). Paradigm version 2 showed no clear evidence of learning - average freezing at recall day 1 was not different than pre-shock freezing. First-lap freezing showed a difference, but this single-lap effect is not useful for many of the neural circuit questions for which this paradigm is meant to facilitate. Also, the claim that mice extinguished first-lap freezing after 1 day is weak. Extinction is determined here by the loss of context discrimination, but this was not strong to begin with. First-lap freezing does not appear to be different between Recall Day 1 and 2, but this analysis was not done.

This is an important point. Following reviewer suggestions, we have replotted our figures for all paradigms to show within-VR freezing (see Supplementary Figures 2, 5 and 6) as the appropriate method for quantifying fear extinction across days. Using an LME model (Tables 16), we quantify freezing during recall days against baseline freezing levels measured before fear conditioning within each VR. In Paradigm 2, while some fear discrimination persists across days, extinction does occur rapidly. After the first lap in the CFC VR, we observed no significant differences in freezing compared to the baseline. These results are shown in the revised Supplementary Figure 5, and the revised text is in lines 393-399.

Paradigm version 3 has some promise, but the magnitude of the context discrimination is modest (~10% difference in freezing). Thus, further optimization of the VR CFC will be needed to achieve robust learning and extinction. This could include factors not thoroughly tested in this study, including context pre-exposure timing and duration and shock intensity and frequency.

We acknowledge that many aspects of this paradigm still need optimization, as virtual reality CFC is in its early stages, and we have not explored all of the parameter space. We describe above the reasoning for this. However, for this revised version of the paper we have added new behavioral data (Supplementary Figure 2 C-E) showing that increasing shock intensities from 0.6 mA to 1 mA enhances freezing, both in the first lap and on average. There are of course many other parameters that are likely important, like the ones pointed out here by the reviewer, but exploring the entire parameter space will take many years and will likely require many labs. The purpose of this paper is to show that VR-CFC fundamentally works and is a starting point from which the field can build on. We have now pointed out in the introduction (lines 54-58) and discussion (lines 730-737, 810-814) that there remains significant scope for improving this paradigm and optimizing parameters in the future.

The second part of the study is a validation of the head-fixed CFC VR protocol through the demonstration that fear conditioning leads to the remapping of dorsal CA1 place fields, similar to that observed in freely moving subjects. The results support this aim and largely replicate previous findings in freely moving subjects. One difference from previous work of note is that VR CFC led to the remapping of the control environment, not just the conditioning context. The authors present several possible explanations for this lack of specificity to the shock context, further underscoring the need for further refinement of the CFC protocol before it can be widely applied. While this experiment examined place cell remapping after fear conditioning, it did not attempt to link neural activity to the learned association or freezing behavior.

This is an interesting observation. We think that the remapping observed in the control context likely occurred due to the absence of reward in a previously rewarded environment. Our prior work has demonstrated that removal of reward causes increased remapping (Krishnan et al., 2022, Krishnan and Sheffield, 2023). In other words, the continued presence of reward within an environment stabilizes CA1 place fields. The Moita et al. (2004) paper, which showed remapping only in the fear conditioned context and not in the control context, provided rats with food pellets throughout the experimental session in both the control and conditioned context— likely to increase exploration necessary for identifying place cells. The presence of reward in the Moita et al experiment could explain the minimal remapping observed in their control context compared to our control context which lacked reward. Another possibility could lie in the differences in the intervals between place cell activity recordings in our study and that of Moita et al. While Moita et al. separated their recordings by just one hour, our recordings were separated by a full day, with a sleep period in between. The absence of sleep and the shorter time interval between conditioning and retrieval sessions in their study could explain the minimal remapping observed by Moita et al. compared to our findings. We have now addressed this discrepancy explicitly in lines 596-606.

Although we agree with the reviewer that it would be informative to perform analysis of how neural activity correlates with freezing responses, we think this warrants its own stand-alone manuscript as the neural dynamics and methods to appropriately analyze them are complicated. We are in the midst of analyzing this data further and will present these findings in a separate publication.

In summary, this is an important study that sets the initial parameters and neuronal validation needed to establish a head-fixed CFC paradigm that produces freezing behaviors. In the discussion, the authors note the limitations of this study, suggest the next steps in refinement, and point to several future directions using this protocol to significantly advance our understanding of the neural circuits of threat-related learning and behavior.
**Reviewer #2 (Public review):**
Summary:In this manuscript, Krishnan et al devised three paradigms to perform contextual fear conditioning in head-fixed mice. Each of the paradigms relied on head-fixed mice running on a treadmill through virtual reality arenas. The authors tested the validity of three versions of the paradigms by using various parameters. As described below, I think there are several issues with the way the paradigms are designed and how the data are interpreted. Moreover, as Paradigm 3 was published previously in a study by the same group, it is unclear to me what this manuscript offers beyond the validations of parameters used for the previous publication. Below, I list my concerns point-by-point, which I believe need to be addressed to strengthen the manuscript.Major comments(1) In the analysis using the LME model (Tables 1 and 2), I am left wondering why the mice had increased freezing across recall days as well as increased generalization (increased freezing to the familiar context, where shock was never delivered). Would the authors expect freezing to decrease across recall days, since repeated exposure to the shock context should drive some extinction? This is complicated by the analysis showing that freeing was increased only on retrieval day 1 when analyzing data from the first lap only. Since reward (e.g., motivation to run) is removed during the conditioning and retrieval tests, I wonder if what the authors are observing is related to decreased motivation to perform the task (mice will just sit, immobile, not necessarily freezing per se). I think that these aspects need to be teased out.

This is an important point and we agree teasing out a lack of motivation versus fearful freezing would be useful. To address the possibility that reduced motivation to run without reward could contribute to the observed freezing behavior, we have now included a no-shock control group in the revised manuscript (n = 7; Supplementary Figure 2A-B, H–I). These control mice experienced the same protocol, including the wearing of a tail coat, but did not receive any shocks. We observed no increases in freezing across days in these controls, confirming that the increased freezing in the Familiar context of our experimental group stems from fear conditioning rather than the removal of reward from a previously rewarded context. If reduced motivation from reward removal were the primary driver, similar freezing patterns would have emerged in the no-shock controls. We have added lines 248-261 in the revised manuscript, discussing this point, and we thank the reviewer for motivating us to do this experiment and analysis.

That said, the precise mechanisms underlying the fear generalization observed in the nonconditioned context—particularly its emergence during later recall days—remain unclear. Studies in freely moving animals have shown that fear memories initially specific to the conditioned context can become generalized with repeated exposures, which may be occurring here (Biedenkapp & Rudy, 2007; Wiltgen & Silva, 2007). Alternatively, it is possible that the combination of fear conditioning and the removal of expected reward contributes to a delayed generalization effect. This may reflect a limitation of our approach, which relies on reward to motivate initial training. As noted by another reviewer, we have now addressed this potential drawback of reward-based training in the discussion (see lines 809-817). Clearly, unique factors specific to the head-fixed VR paradigm may contribute to this phenomenon. Understanding the mechanisms underlying fear generalization in the head-fixed VR CFC paradigm will be a valuable direction for future research.

(2) Related to point 1, the authors actually point out that these changes could be due to the loss of the water reward. So, in line 304, is it appropriate to call this freezing? I think it will be very important for the authors to exactly define and delineate what they consider as freezing in this task, versus mice just simply sitting around, immobile, and taking a break from performing the task when they realize there is no reward at the end.

As noted in point 1 above, we have added a no-shock control group (n = 7; Supplementary Figure 2A-B, H–I) to determine whether the observed freezing was driven by fear conditioning or by reduced motivation to run in the absence of reward. The absence of increased freezing in these controls supports the interpretation that the behavior in the conditioned group is fearrelated. In future studies, incorporating additional physiological measures—such as heart rate monitoring—could further help distinguish fear-related freezing from other forms of immobility.

(3) In the second paradigm, mice are exposed to both novel and (at the time before conditioning) neutral environments just before fear conditioning. There is a big chance that the mice are 'linking' the memories (Cai et al 2016) of the two contexts such that there is no difference in freezing in the shock context compared to the neutral context, which is what the authors observe (Lines 333-335). The experiment should be repeated such that exposure to the contexts does not occur on the conditioning day.

This is an interesting idea. However, if memory linking were driving the observed freezing patterns, we would expect to see similarly reduced fear discrimination across all three paradigms, as mice experience both contexts sequentially in each case. However, this effect appears to be specific to Paradigm 2, suggesting this may be due to other factors. We agree it would be informative to eliminate pre-conditioning exposure to both environments—to assess whether this improves fear discrimination and helps clarify the potential contribution of memory linking. This is something we plan to do in future studies that are beyond the scope of this initial paper on VR-CFC.

(4) On lines 360-361, the authors conclude that extinction happens rapidly, within the first lap of the VR trial. To my understanding, that would mean that extinction would happen within the first 5-10 seconds of the test (according to Figure S1E). That seems far too fast for extinction to occur, as this never occurs in freely behaving mice this quickly.

We agree with the reviewer that extinction in Paradigm 2 appears to occur relatively rapidly.

However, the average time to complete the first lap in the fear-conditioned context in Paradigm 2 is 25.68 ± 5.55 seconds (as stated in line 384), indicating that extinction occurs within approximately the first 30 seconds of context exposure—not within 5–10 seconds. This is specific to Paradigm 2 and does not happen in either of the other paradigms, as shown in Supplementary Figure 4. For clarification, Figure S1E pertains to baseline running in Paradigm 1 and does not apply to Paradigm 2.

As the reviewer points out, even at 30 seconds, extinction seems to be happening more quickly in Paradigm 2 than seen in freely moving setups. This may be due to a key structural difference in our setup. The VR-CFC task is organized into discrete trials, with mice being teleported back to the start after reaching the end of the virtual track. Completing a full lap without receiving a shock could serve as a clear signal that the threat is no longer present within the environment as the completion of a lap means that the animals have surveyed all locations within the environment. This structure could accelerate extinction compared to freely moving setups, where animals take longer to explore their complete environment due to the lack of discrete trials. Although this is true for all our paradigms, the accelerated extinction seen in paradigm 2 versus 1 and 3 may be driven by other factors. As noted by the reviewers, other task parameters—such as context pre-exposure timing, shock intensity, and conditioning duration— are likely to play a role in shaping extinction dynamics. These factors warrant further investigation, and we plan to explore them in future studies to better understand the conditions influencing extinction in the VR-CFC paradigm.

(5) Throughout the different paradigms, the authors are using different shock intensities. This can lead to differences in fear memory encoding as well as in levels of fear memory generalization. I don't think that comparisons can be made across the different paradigms as too many variables (including shock intensity - 0.5/0.6mA can be very different from 1.0 mA) are different. How can the authors pinpoint which works best? Indeed, they find Paradigm 3 'works' better than Paradigm 2 because mice discriminate better between the neutral and shock contexts. This can definitely be driven by decreased generalization from using a 0.6mA shock in Paradigm 3 compared to 1.0 mA shock in Paradigm 2.

The reviewer brings up important points here. We have now added new data evaluating 0.6 mA shocks in Paradigm 1 (Supplementary Figure 2A–E, n=8). These data show that 1.0 mA shocks produced stronger conditioned responses and greater fear discrimination compared to 0.6 mA. Our goal in Paradigm 3 was to begin with a lower shock intensity and assess whether additional modifications—specifically the shorter ISI and retention of the tail-coat during recall—could enhance fear conditioning. Surprisingly, despite the weaker shock intensity, Paradigm 3 resulted in improved discrimination and freezing behavior relative to Paradigm 2. We have now clarified this point in the manuscript (lines 466-470), and we interpret this outcome as evidence that the shorter ISIs and contextual cue continuity (tail-coat) likely play a more significant role in enhancing learning and recall. However, as noted in the text (lines 511-514), further testing is needed to determine the individual contributions of each parameter to successful VR-CFC. Fully optimizing the parameter settings will take additional time and resources, and we aim to continually refine the parameter space in the future, as has been done over the years for freely moving animals.

(6) There are some differences in the calcium imaging dataset compared to other studies, and the authors should perform additional testing to determine why. This will be integral to validating their head-fixed paradigm(s) and showing they are useful for modeling circuit dynamics/behaviors observed in freely behaving mice. Moreover, the sample size (number of mice) seems low.

The one notable difference between our imaging study and that done in freely moving animals is that we observed remapping of place cells in the control context. In contrast, Moita et al. (2004) reported more stable place fields in the control context. A key distinction is that their study included rewards in the control context, which may have contributed to the spatial stability. We now discuss this difference in the manuscript (lines 599-605).

It should be noted that there are many key distinctions among paradigms that study neural activity during fear conditioning in freely moving animals. These include varying exposure times to environments (1–6 days), the time interval between neural activity recordings, and the use of food rewards during the experiment stages in freely moving animals to encourage exploration for place cell identification. Although freely moving paradigms that investigate fear conditioning and place cells are heterogeneous, we were encouraged by the replication of several key findings. This validates VR-based CFC as a viable tool for neural circuit investigations. While future work will include more thorough analyses, our current findings demonstrate the paradigm's effectiveness for modeling circuit dynamics and behavior. We have now expanded our dataset, which includes four additional mice, further corroborating these original findings.

(7) It appears that the authors have already published a paper using Paradigm 3 (Ratigan et al 2023). If they already found a paradigm that is published and works, it is unclear to me what the current manuscript offers beyond that initial manuscript.

The reviewer is correct that we have published a paper using Paradigm 3. However, this manuscript goes beyond that one and provides a much more comprehensive description and fundamental analysis of the behavior and experimental parameters regarding VR-CFC, allowing the research community to adapt our paradigm reproducibly. While Ratigan et al. (2023) offered only a minimal description of behavior and included just Paradigm 3, we present two additional paradigms along with neuronal validation using hippocampal place cells. We have now explicitly stated this in the introduction (lines 50-55).

(8) As written, the manuscript is really difficult to follow with the averages and standard error reported throughout the text. This reporting in the text occurred heterogeneously throughout the text, as sometimes it was reported and other times it was not. Cleaning this reporting up throughout the paper would greatly improve the flow of the text and qualitative description of the results.

We completely agree with this point and have now cleaned up the text, leaving details only in a few places we felt were important.

**Reviewer #3 (Public review):**
Summary:Krishnan et al. present a novel contextual fear conditioning (CFC) paradigm using a virtual reality (VR) apparatus to evaluate whether conditioned context-induced freezing can be elicited in head-fixed mice. By combining this approach with two-photon imaging, the authors aim to provide high-resolution insights into the neural mechanisms underlying learning, memory, and fear. Their experiments demonstrate that head-fixed mice can discriminate between threat and non-threat contexts, exhibit fear-related behavior in VR, and show context-dependent variability during extinction. Supplemental analyses further explore alternative behaviors and the influence of experimental parameters, while hippocampal neuron remapping is tracked throughout the experiments, showcasing the paradigm's potential for studying memory formation and extinction processes.Strengths:Methodological Innovation: The integration of a VR-based CFC paradigm with real-time twophoton imaging offers a powerful, high-resolution tool for investigating the neural circuits underlying fear, learning, and memory.Versatility and Utility: The paradigm provides a controlled and reproducible environment for studying contextual fear learning, addressing challenges associated with freely moving paradigms.Potential for Broader Applications: By demonstrating hippocampal neuron remapping during fear learning and extinction, the study highlights the paradigm's utility for exploring memory dynamics, providing a strong foundation for future studies in behavioral neuroscience.Comprehensive Data Presentation: The inclusion of supplemental figures and behavioral analyses (e.g., licking behaviors and variability in extinction) strengthens the manuscript by addressing additional dimensions of the experimental outcomes.Weaknesses:Characterization of Freezing Behavior: The evidence supporting freezing behavior as the primary defensive response in VR is unclear. Supplementary videos suggest the observed behaviors may include avoidance-like actions (e.g., backing away or stopping locomotion) rather than true freezing. Additional physiological measurements, such as EMG or heart rate, are necessary to substantiate the claim that freezing is elicited in the paradigm.

To strengthen our claim that freezing is a conditioned response in this task, we have taken three key steps:

(1) We adjusted our freezing detection threshold from 1 cm/s to near 0 cm/s to capture only periods where the animal is virtually motionless on the treadmill. We validated this approach in Figure 2, particularly in the zoomed-in track position trace in Figure 2A, which clearly shows that the identified freezing epochs correspond to no change in track position. All analyses and figures have been updated to reflect this more stringent threshold.

(2) We have added a no-shock control group in the revised manuscript (n = 7; Supplementary Figure 2A-B, H–I) where mice experienced the same protocol, including wearing a tail-coat, but received no shocks. These mice showed no increases in freezing behavior, which further demonstrates that the increased freezing we observe is a result of fear conditioning.

(3) We have added a new supplementary video (Supplementary Video 2) that better illustrates the freezing behavior in our task.

That said, we fully agree with the reviewer that freezing is not the only defensive response observed. Other behaviors—such as hesitation, backward movement, and slowing down—also emerge that are unique to our treadmill-based paradigm. We chose to focus on freezing in this manuscript to align with convention in freely moving fear conditioning studies and to facilitate direct comparisons. We agree that additional physiological measurements (e.g., EMG or heart rate) would provide further validation and could help distinguish between different forms of defensive responses. We view this as an important future direction and plan to incorporate such measures in upcoming studies. We highlight this in the results section (lines 175-179, 262-268) and in the discussion (lines 739-750).

Analysis of Extinction: Extinction dynamics are only analyzed through between-group comparisons within each Recall day, without addressing within-group changes in behavior across days. Statistical comparisons within groups would provide a more robust demonstration of extinction processes.

This is an important distinction and we have now added figures (Supplementary Figures 2H-I, 5C-D, 6C-D) showing within-VR behavior across Recall days, along with statistical comparisons and a description of the extinction process based on these results.

Low Sample Sizes: Paradigm 1 includes conditions with very low sample sizes (N=1-3), limiting the reliability of statistical comparisons regarding the effects of shock number and intensity.

Increasing sample sizes or excluding data from mice that do not match the conditions used in Paradigms 2 and 3 would improve the rigor of the analysis.

While we included all conditions in Figure 2 for completeness, we have separated these conditions in Supplementary Figure 2 to ensure clarity. This allows researchers interested in this paradigm to see the approximate range of conditioned responses observed across different parameters. When comparing Paradigm 1 with Paradigms 2 and 3, we have only used data from 1mA, 6 shocks condition.

Potential Confound of Water Reward: The authors critique the use of reward in conjunction with fear conditioning in prior studies but do not fully address the potential confound introduced by using water reward during the training phase in their own paradigm.

We agree this is a point that needs discussion. We have now noted the limitation of using water rewards during training in the discussion section, particularly its effect on the animal’s motivation in the long term and on place cell activity (lines 814-820).

**Recommendations for the authors**

**Reviewer #1 (Recommendations for the authors):**
I suggest changing "3 paradigms" to "3 versions of a CFC paradigm," as the paradigm is fundamentally the same, but parameters were adjusted towards finding an optimal protocol.

We have changed this phrasing where applicable.

Figure S2: There appear to be different sets of shock parameters for different mice, most with an n of 1 or 2. This is not reliable for making a decision for optimal shock parameters and should not be discussed in that way until a full-powered comparison is completed. Also, the N adds up to 19, yet only 18 are described as being included in the study.

We thank the reviewer for this important point. We agree that the current study is not powered to definitively identify optimal parameter settings. We have been careful not to interpret it in that way in the text. Rather, we adopted a commonly used starting point from the freely moving literature—1 mA with six shocks—as our initial condition (lines 196-199). To provide context for others interested in pursuing this work, we have presented a range of conditioned responses from different parameter combinations to illustrate potential variability. In most cases, these data are intended for illustrative purposes only and are not meant to support firm conclusions. We agree that a systematic and fully powered investigation of each parameter would be highly valuable, and we plan to pursue this in future work (and hope other labs contribute to this goal, too), much like the iterative optimizations performed in freely moving paradigms over time.

We thank the reviewer for catching the sample size discrepancy and have now corrected it.

The number of animals for the no-shock condition should be included.

Thank you. We have now included this.

A possible explanation for the lower fear and poorer discrimination in versions 2 and 3 could be that 10 min pre-exposure to the CFC context on day -1 led to latent inhibition. Shorter (or eliminated) pre-exposure may improve outcomes.

We agree that the exposure time is a parameter that we should explore. We have highlighted this in the discussion (lines 729-736) as a parameter that is worth testing in the future.

For analysis of extinction, it is best to establish this within condition - is freezing to the CFC context significantly reduced compared with initial recall and similar to pre-training freezing? By using discrimination as your index of extinction, increases in control context freezing/inactivity can eliminate context discrimination without the conditioned response of freezing actually undergoing extinction.

This is a good point, and we have now included analysis and conclusions based on a within-VR comparison for the analysis of fear extinction (Supplementary Figures 2H-I, 5C-D, 6C-D).

**Reviewer #3 (Recommendations for the authors):**
Clarification of Treadmill Shape: The manuscript describes the treadmill as "spherical" throughout. However, based on representative images and videos, the treadmill appears cylindrical. This discrepancy should be clarified to ensure consistency between the text and visuals.

The reviewer is correct that the treadmill is cylindrical, and this was an error on our part. We have corrected it throughout.

Figure and Legend Labeling: To improve clarity, all figures and their legends should be explicitly labeled with the corresponding paradigm (1, 2, or 3) to facilitate interpretation.

We have now added a label on all figures that clarifies which Paradigm the figures are referring to. We have also explicitly added this to the figure legends.

Objective Language: Subjective language, such as "since we wanted animals to" (Line 850), should be revised to reflect an objective tone (e.g., "to allow animals to"). Similarly, phrases like "We believe" (Line 896) should be avoided to maintain an unbiased presentation.

We have removed subjective language from our text.

Placement of Future Directions: Speculations on future experimental plans, such as the use of sex as a biological variable (Lines 895-903), should be included in the Discussion section rather than the Methods. Additionally, remarks about the responsiveness of female mice to tail shocks should be moved to the main text for proper contextualization.

We have moved these lines as suggested by the reviewer.